# Evaluation of long term Northern Hemisphere snow water equivalent products

Colleen Mortimer[1], Lawrence Mudryk[1], Chris Derksen[1], Kari Luojus[2], Ross Brown[1], Richard Kelly[3], and Marco Tedesco[4]

[1]Climate Research Division, Environment and Climate Change Canada, Toronto, Canada
[2]Finnish Meteorological Institute, Helsinki, Finland
[3]Department of Geography and Environmental Management, University of Waterloo, Canada
[4]Lamont Doherty Earth Observatory, Columbia University, Palisades NY; NASA Goddard Institute for Space Studies, New York, USA

*Correspondence to*: Colleen Mortimer (colleen.mortimer@canada.ca)

**Abstract.** Nine gridded northern hemisphere snow water equivalent (SWE) products were evaluated as part of the European Space Agency (ESA) Satellite Snow Product Intercomparison and Evaluation Exercise (SnowPEx). Three categories of datasets were assessed: (1) those utilizing some form of reanalysis (the NASA Global Land Data Assimilation System version 2 – GLDAS-2; the European Centre for Medium-Range Weather Forecasts (ECMWF) interim land surface reanalysis – ERA-Interim/Land and ERA5; the NASA Modern-Era Retrospective Analysis for Research and Applications version 1 (MERRA) and version 2 (MERRA2); the Crocus snow model driven by ERA-Interim meteorology – Crocus); (2) passive microwave remote sensing combined with daily surface snow depth observations (ESA GlobSnow v2.0); and (3) standalone passive microwave retrievals (NASA AMSR-E SWE versions 1.0 and 2.0) which do not utilize surface snow observations. Evaluation included validation against independent snow course measurements from Russia, Finland, and Canada, and product intercomparison through the calculation of spatial and temporal correlations in SWE anomalies. The standalone passive microwave SWE products (AMSR-E v1.0 and v2.0 SWE) exhibit low spatial and temporal correlations to other products, and RMSE nearly double the best performing product. Constraining passive microwave retrievals with surface observations (GlobSnow) provides comparable performance to the reanalysis-based products; RMSE over Finland and Russia for all but the AMSR-E products is ~50 mm or less, with the exception of ERA-Interim/Land over Russia. Using a seven-dataset ensemble that excluded the standalone passive microwave products reduced the RMSE by 10 mm (20%) and increased the correlation from 0.67 to 0.78 compared to any individual product. The overall performance of the best multi-product combinations is still at the margins of acceptable uncertainty for scientific and operational requirements; only through combined and integrated improvements in remote sensing, modeling, and observations will real progress in SWE product development be achieved.

# 1 Introduction

Temporally (~20–30 years) and spatially (~10–20 km) consistent estimates of daily SWE over seasonal snow covered land are required for many applications including climate model evaluation (Mudryk et al., 2018a), verification of seasonal forecasts (Sospedra-Alfonso et al., 2016), annual updates to climate assessments (e.g. Mudryk et al., 2018b; 2019), and determination of freshwater availability (Barnett et al. 2005; Clark et al. 2011). There is a growing number of gridded SWE datasets available to the snow community, but these are typically affected by one or more critical shortcomings related to:

*1. Challenges in using point measurements:* Meaningful spatially continuous information can be derived from surface observations for regions and time periods with a sufficiently dense observing network (Dyer and Mote, 2006; Brown and Derksen, 2013); as an alternative to snow depth, snowfall measurements can also be integrated (Broxton et al., 2016). However, both snow depth and snowfall measurements from single point locations are intrinsically limited by a lack of confidence in how they capture the landscape mean across coarse grid cells (Meromy et al., 2013), which is particularly problematic in areas of mixed forest vegetation, open areas prone to wind redistribution, and complex topography (most snow covered regions fall into at least one of these categories). Furthermore, there remain expansive alpine and northern regions with insufficient coverage by conventional observing networks (Brown et al., 2019).

*2. Reliance on models driven by atmospheric reanalysis:* Most modern reanalysis products include output of land surface variables such as SWE (Balsamo et al., 2015; Gelaro et al., 2017); alternatively the meteorology from these datasets can be used to force snow models (Brown et al., 2003; Brun et al., 2013). While these snow schemes are of varying complexity, they typically do not account for important processes such as snow-vegetation interactions and redistribution by blowing snow. In addition, the spread in SWE estimates among differing reanalyses is large: not only do differences between snow models introduce uncertainties (Mudryk et al., 2015), but model-based approaches are also sensitive to the precipitation forcing, which itself is challenging to validate in complex terrain and observation sparse regions (Lundquist et al., 2015; Henn et al., 2018). There may also be temporal inconsistencies in the forcing data related to changes in the observational streams assimilated in the reanalyses (Robertson et al., 2011).

*3. Coarse spatial resolution:* Whether derived from passive microwave satellite measurements or some form of model reanalysis, the typical resolution of existing gridded SWE datasets is 25 to 100 km. While synoptic scale patterns can be resolved at this resolution, spatial variability in SWE due to topographic and land cover heterogeneity is not adequately captured. Coarse resolution is a particularly critical limitation in alpine regions, which are masked out completely in some products (e.g. Takala et al., 2011). While this is a reasonable decision for some coarse resolution products, it nevertheless is a source of frustration for users. Coarse resolution also makes validation of SWE products challenging: the validation of large grid cells with single point measurements is conceptually unsatisfying and statistically non-robust. Regional climate models can provide higher resolution SWE information, but the computational cost related to complex atmospheric physics schemes

is, at least at present, a limiting factor in producing long time series (Wrzesien et al., 2018). There may be potential for cross-polarized C-band SAR to provide high spatial resolution snow depth information in mountain areas (Lievens et al., 2019), but these estimates currently lack a physical explanation. Since cross-polarized C-band SAR data are only available since the launch of Sentinel-1A in 2014, there is limited potential to provide climate-relevant time series.

*4. Inability of remote sensing data to constrain uncertainty:* The number of purely satellite derived SWE datasets is limited, and uncertainty in standalone passive microwave retrievals can be high (Kelly et al., 2003). The combination of passive microwave and surface snow depth measurements (within the GlobSnow product; Takala et al., 2011) was shown to yield performance similar to snow models driven by atmospheric reanalysis (Mudryk et al., 2015), but it relies heavily on background fields and constraints generated from re-gridded surface snow depth observations (Pulliainen, 2006). The microwave remote sensing community has made great progress in understanding and quantifying error sources (snow microstructure, deep snow, wet snow, vegetation, lake ice), all of which are exacerbated by the coarse resolution of passive microwave measurements (Foster et al., 2005; Durand et al., 2011; Lemmetyinen et al., 2011; Durand and Liu, 2012).

Previous studies have demonstrated the potential for using multi-product SWE ensembles in order to improve estimates of observed snow-related quantities (e.g. SWE and snow cover fraction, integrated snow mass, snow cover extent and trends in these quantities) and to constrain uncertainty (Mudryk et al., 2015, 2017, 2018a; Krinner et al., 2018). The intent in such a strategy is that uncorrelated errors between products of the same type average out, so the limitations and shortcomings of a given class of products offset one another. Ideally, such ensembles would draw from as many types of products as possible and use multiple versions of each type of product. To date, these ensembles have relied heavily on models driven by atmospheric reanalyses and include only a single dataset (GlobSnow) that utilizes remote sensing. While SWE or snow depth products can be derived using InSAR techniques (Deeb et al., 2011) and airborne LiDAR data (Painter et al., 2016), such products are only available for regionally and temporal limited domains. Hence, the long time series of passive microwave measurements provide the most straightforward pathway to increase the use of satellite data within observational SWE ensembles. Before existing passive microwave derived SWE products can be included, however, an assessment is needed because of markedly different climatological patterns (Fig. 1; discussed further in Sect. 3.1). The specific objectives of this study are to evaluate gridded Northern Hemisphere SWE products by (1) *validation* with independent surface observations, and (2) *intercomparison* through calculation of the spatial and temporal correlations in SWE anomalies.

## 2 Datasets and methods

### 2.1 Gridded SWE products

We evaluate three categories of northern hemisphere gridded SWE products: (1) standalone passive microwave retrievals (AMSR-E SWE v1.0 and v2.0); (2) passive microwave estimates combined with surface snow depth observations (GlobSnow

v2.0), and (3); products which utilize some form of reanalysis (Crocus, GLDAS-2, ERA-Interim/Land, ERA5, MERRA, MERRA2). A summary of these nine SWE datasets is provided in Table 1. All the products provide SWE directly and are available at daily or sub-daily frequency. For the four products available at sub-daily frequency, we either obtained daily mean versions directly from the product's distribution site (MERRA, MERRA2) or sampled a consistent sub-daily snapshot for each calendar day (ERA-Interim/Land, ERA-5) which we consider to be representative of the daily mean value. The analyses described subsequently in Section 2.3 were conducted for the period 2002 –2010 to maximize temporal overlap between products.

1. *Standalone passive microwave*: The NASA AMSR-E SWE v1.0 product (https://nsidc.org/data/AE_DySno/versions/2, Tedesco et al., 2004) is described in Kelly (2009) and evaluated in Tedesco and Narvekar (2010). Brightness temperature thresholds are utilized to identify shallow and non-shallow dry snow areas, with the depth of shallow snow set to 5 cm (Kelly et al., 2003). SWE is retrieved based on a brightness temperature difference approach (37-19 GHz; based on the original formulation of Chang et al. (1990)) with enhancements to account for the influence of vegetation, to address deeper snowpacks (through the use of 10 GHz measurements), and to consider the dynamic influence of snow grain size (based on the assumption that as snow depth increases, the depth average grain size increases). Snow depth is converted to SWE using the snow climate classification of Sturm et al. (1995) and snow density climatologies from Brown and Braaten (1998) and Krenke (2004). Building on the v1.0 AMSR-E SWE product, NASA's current v2.0 AMSR-E SWE algorithm utilizes an artificial neural network, snow emission modelling, and climatological snow depth data for the estimation of snow depth, and the detection of dry versus wet snow conditions (Tedesco and Jeyaratnam 2016). Snow density maps based on Sturm et al. (2010) are employed for conversion of retrieved snow depth to SWE. Unlike the GlobSnow approach described next, both NASA AMSR-E SWE algorithms are self-contained and do not rely on any external temporally variable snow measurements.

2. *Synergistic passive microwave + in situ:* The European Space Agency GlobSnow v2.0 SWE product (data available at www.globsnow.info) is based on a retrieval method first described in Pulliainen (2006). The approach evolved from standalone passive microwave algorithms (so it also relies on 19 and 37 GHz measurements), but the retrieval also integrates daily surface snow depth measurements. First, daily climate station snow depth observations are kriged to form a continuous background field independent of passive microwave retrievals. This first guess snow depth field is used as input to two iterations of forward microwave emission model simulations, one to estimate grain size, the second to estimate snow depth (Takala et al., 2011). A temporally and spatially fixed snow density value of 0.24 g cm$^{-3}$ is applied to convert snow depth to SWE. Alpine areas are excluded due to the known limitations of this technique in regions with complex sub-grid topographical heterogeneity (Takala et al., 2011).

3. *Land surface models and reanalysis*: Six SWE datasets derived from combinations of models driven by reanalysis meteorology were used for comparison with the passive microwave products: the NASA Global Land Data

Assimilation System version 2 – GLDAS-2; the European Centre for Medium-Range Weather Forecasts (ECMWF) interim land surface reanalysis – ERA-Interim/Land; and ECMWF ReAnalysis version 5 - ERA5; the NASA Modern-Era Retrospective Analysis for Research and Applications, version1 – MERRA, and version2 - MERRA2; the Crocus snow model driven by ERA-Interim meteorology – Crocus. We refer to these datasets as *snow analyses*. It is important to note that spread among the snow analyses does not only depend on differences in the forcing data; in fact, a substantial portion of the spread stems from differences in the complexity and parametrizations of their respective snow schemes (see Mudryk et al., 2015). For example, both Crocus and ERA-Interim/Land use the same forcing data but employ different land models with different snow schemes which yield significantly different validation results (Section 3.2). The impact of snow depth observations also differs between reanalysis products. Snow depth observations are directly assimilated into ERA5. For ERA-Interim/Land, however, only the forcing meteorology includes explicit assimilation of point snow depth measurements (the SWE produced by ERA-Land does not). Therefore, for ERA-Interim/Land, the use of snow depth information is one step removed from the final SWE estimates compared to ERA5, although the assimilation of snow information impacts variables such as lower tropospheric temperatures which obviously have an indirect impact on snow.

## 2.2 Snow course data

The suite of gridded SWE products described in Section 2.1 are validated with a network of in situ snow course measurements from multiple national and regional agencies. These data consist of manual gravimetric snow measurements made at multiple locations along a pre-defined transect that are averaged to obtain a single SWE value for a given snow course on a given day. Measurements are collected along the same transect multiple times each snow season. By averaging multiple samples along a transect, the resulting SWE measurement provides better representation of sub-grid scale variability than a single point measurement and so is more suitable for evaluation of SWE at the scale of the gridded products. These snow course data are fully independent of the point snow depth measurements assimilated into GlobSnow and ERA5. Transect length, number of samples collected along each transect, and sample aggregation methods differ among reporting agencies as described below.

Russia has a long-term snow course network located near 517 meteorological stations (Bulygina et al., 2011). The snow survey transects extend for 1 to 2 km in open areas, and 500 m at forested sites. Measurements are made every ten days when at least half of the visible area around a station is snow-covered, except at forested sites where measurements are made once per month prior to 20 January. Sampling frequency is increased to five days during the spring snow melt season. The Finnish snow course network, maintained by the Finnish Environment Institute (SYKE), consists of approximately 200 transects distributed across the country. Measurements are conducted monthly around the 15[th] of each month, with a subset of snow courses also measured at the end of each month. Each snow course is 2 to 4 km long and extends through variable land cover consistent with the surrounding landscape (Haberkorn, 2019).

The Canadian snow course data are a recently updated collection pooled from a series of national and regional networks described in Brown et al. (2019). There is no comprehensive national strategy in Canada to obtain a spatially representative collection of snow course measurements. Snow courses are maintained by various jurisdictions resulting in a spatially heterogeneous sample distribution heavily biased towards population centres. For 2002 through 2010, there were more than 1000 unique snow course locations across Canada with varying sampling frequency. Measurements are typically made around the 1$^{st}$ and 15$^{th}$ of each month during the snow season (November to April). Snow courses are roughly 150 to 300 m long consisting of five to ten sampling locations (Brown et al., 2019). While the network density is sparse across Canada and the transects are shorter in length than the Russian and Finnish data, previous analysis suggests the measurements still capture reasonable landscape mean values (Neumann et al., 2006).

Because snow course measurements are only acquired during the snow season and zero SWE values are not reported in a consistent manner across all jurisdictions, zero SWE is not a reliable measure of snow-free conditions. All zero snow course observations were therefore removed prior to spatiotemporal aggregation (Sect. 2.3); SWE product zero values were also excluded. Finally, it is difficult to attach specific uncertainty values to the snow survey measurements because non-standard sampling tools are used between snow courses (e.g. no consistent snow corer diameter). Full discussion of snow course measurement protocols and instruments is available elsewhere (Goodison et al., 1981; Brown et al., 2019; Haberkorn et al., 2019), but there is no doubt that uncertainty associated with the individual measurements (± approximately 5%; Brown et al., 2019) is overwhelmed by uncertainty in how the snow course measurements represent the landscape mean at the scale of the gridded SWE products.

## 2.3 Validation and intercomparison methods

We assess the gridded products in two separate analyses conducted for the snow season (defined here as November – April (NDJFMA)) from November 2002 – April 2010. The first assessment is termed a *validation* because it evaluates each gridded product using the snow course data as a measure of ground truth. While the relative sparseness of the snow course measurements limits the assessment's spatial and temporal completeness, it nonetheless considers a broad range of snow classes covering both Northern Hemisphere continents (Fig. 2) and considers seasonal variability from November through April over eight years of interannual variability. We are unaware of any other studies that have evaluated the breadth of products examined here with similarly representative data and with comparable spatial and temporal coverage. The second assessment is termed an *intercomparison* and is similar to the analysis performed in Mudryk et al. (2015). This second type of analysis is spatially and temporally complete (across the seasons and period considered). We use this analysis as it provides a more complete measure of differences among the products and is able to more readily discern differences and discontinuities between products than the validation analysis (see results regarding ERA5 in Sect. 3.3).

For the *validation* analysis, SWE product grid cells must be matched in both space and time with the snow course measurements. To achieve this, snow course observations from Canada and Finland were first grouped into bi-weekly periods using a 16 day window centred on the 1st or 15th of each month. Likewise, over Russia, observations were grouped into ten-day periods centred on the typical measurement dates (10th, 20th, 30th of each month). For each temporal grouping, snow course measurements falling within a given 25 x 25 km EASE grid cell (Brodzik et al., 2012) were averaged together, thereby forming a gridded snow course field (Fig. 2). Roughly 30% of these snow course grid cells had two or more separate snow courses which were averaged together while the remaining 70% had only one snow course observation. Grouping the snow course data had the largest impact over Canada and Russia where 35% and 20% of grid cells, respectively, had multiple snow courses. Although Finland's snow course network is representative of the landscape's different snow-climate classes(Sturm et al., 1995), in Canada, and to a lesser extent over Russia, tundra environments which are often remote, are under-sampled while maritime and alpine snow types are oversampled (Fig. 2).

For the validation analysis, we included all nine products in Table 1, to consider the range of available products and show the difference in performance between subsequent product generations (e.g. MERRA to MERRA2). For a given measurement date, each EASE grid cell with snow course data was paired with corresponding SWE values from each of the nine gridded products. The paired SWE values correspond to the grid cell at each product's native resolution that intersects with the centroid of the snow course EASE grid cell. In order to fairly compare how the gridded products perform against one another, only snow course data from EASE grid cells with corresponding paired values from all nine of the SWE products were analysed. This means that regions of complex topography are implicitly excluded from the validation analysis because they are masked in GlobSnow. Analyses were conducted for the snow season only (November – April). Bias and root mean squared error (RMSE) were calculated for each product-snow course pair and then averaged over the full November 2002 – April 2010 time period; correlation was calculated from all data pairs for the November 2002 – April 2010 period. To understand the influence of seasonality on product performance, bias, RMSE and correlation were also computed across all years for each biweekly period (10 day period for Russia). Validation statistics were calculated separately for each national snow course dataset in order to separate any sensitivity to differences in snow course measurement protocol and sample distribution. Finally, to determine the influence of SWE magnitude on product performance, all snow course-product SWE pairs were binned into 10 mm increments according to the snow course SWE. For each 10 mm increment the average product SWE was plotted against the bin midpoint.

The *intercomparison* analysis does not consider the snow course measurements, only the nine gridded SWE products. For this analysis, daily SWE from each product was interpolated to a regular 1° x 1° longitude–latitude grid. SWE values over glaciers and large lakes were excluded based on the MERRA land fraction mask (consistent with Mudryk et al., 2015). To determine the strength of agreement among datasets we use three metrics, all applied to SWE or snow mass anomalies (i.e. with the seasonal cycle removed). We only consider anomalies due to the results from Mudryk et al. (2015), which demonstrated that while different snow products can have substantial spread in their climatological snow estimates, one can and should expect a

reasonable degree of agreement in their interannual and intraseasonal variability. First, we considered the correlation between each product's time series of daily Northern Hemisphere snow mass anomalies (SWE integrated over the entire Northern Hemisphere land area). Each product's time series was calculated using its respective climatology (determined for the snow season over the November 2002–April 2010 period). A correlation coefficient was calculated for each pair of datasets by correlating the two snow mass anomaly time series cropped to the snow season (November–April) over the April 2002 – November 2010 period. Secondly, we considered correlations between the patterns of anomalous SWE fields. Daily SWE anomalies were calculated for each product using its respective climatology. For each dataset pair, we calculated the daily pattern correlation between the two anomalous SWE fields and averaged the sequence of correlation values over the snow season for the 2002–2010 period. These first two metrics are bulk measures of agreement, specifically in their estimates of Northern Hemisphere snow mass anomalies and the average agreement of their pattern correlations. Finally, we also considered 'local' correlation maps of anomalous SWE. As above, we calculated daily anomalous SWE fields. Then for each dataset pair, we calculated the correlation coefficient between the daily time series of anomalous SWE at each location on the 1° x 1° grid. The correlation calculation only considers the snow season (November – April) over the November 2002 – April 2010 period. This third metric allows us to consider which regions agree more and less among the various products.

## 3 Results

### 3.1 Climatology

There is notable disagreement in the climatological SWE distribution over the Northern Hemisphere land area between standalone passive microwave products and the other data sources (Fig. 1). The pattern of high and low SWE between western and eastern Siberia is reversed for the snow analyses and GlobSnow versus the two AMSR-E algorithms. This inconsistency across Eurasia was also identified in analysis of older versions of passive microwave derived SWE data (e.g. Rawlins et al., 2007), reanalysis, and climate model simulations (see Fig. 2 in Clifford et al., 2010). The AMSR-E products also fail to capture a pronounced region of high SWE in eastern Canada present in the other datasets. The GlobSnow climatology is in close agreement with the snow analyses, particularly over Eurasia. The snow analyses and GlobSnow also agree with other SWE climatologies derived from other sources covering different time periods and thus not included in this study (see Brown and Mote, 2009; Liston and Hiemstra, 2011).

The difference in climatological SWE patterns is not solely due to the well-documented systematic underestimation in passive microwave retrievals when SWE exceeds 150 mm (Markus et al., 2006). Eastern Siberia is a low winter-season precipitation environment with very cold surface temperatures. These are ideal conditions for a thin, low density snowpack (see Liston and Hiemstra, 2011), likely composed primarily of faceted snow grains due to kinetic metamorphism, as seen in the Canadian Arctic (Derksen et al., 2014) and Alaskan North Slope (Hall, 1987). Thin snow composed of large faceted grains results in

exaggerated scattering relative to the amount of SWE (Hall et al., 1991), hence the comparatively large SWE estimates for the standalone passive microwave products.

The reason the standalone passive microwave products fail to capture higher SWE in western Siberia, Russia, northern Europe, and eastern Canada is less clear, but may be related to weaker scattering signatures from smaller grained and deeper snow,
which is further masked by microwave emission from forest cover. The ability of GlobSnow to retain sensitivity to deeper snow than the AMSR-E products is due to the assimilation of daily surface snow depth observations which work to 'nudge' the retrievals to higher values (Pulliainen, 2006). In observation-sparse regions such as northern Quebec, the GlobSnow estimates are more heavily weighted to the passive microwave retrievals, which increases uncertainty in these areas (Larue et al., 2017; Brown et al., 2018) compared to forested, deep snow regions with a dense observation network such as Finland
(Takala et al., 2011).

### 3.2 Comparison with surface measurements

The nine gridded SWE datasets were compared to Canadian, Finnish, and Russian snow course measurements for all snow seasons (November–April) over the November 2002 – April 2010 period. A summary of the validation results is provided in Fig. 3.

All products exhibit weaker skill over Canada, where the RMSE for all products is roughly twice that of Finland and Russia. Larger absolute bias and RMSE over Canada may be attributed, in part, to a higher average SWE since the mean SWE of all snow course grid cells used for validation (Sect. 2.3) is 143 mm in Canada compared to 96 mm in Finland and 76 mm in Russia. However, the RMSE, expressed as a percentage of the mean observed SWE (of grid cells used in the analysis) is still higher over Canada for almost all products, indicative of poorer relative performance. The exception is ERA-Interim/Land
which has poorer relative performance over Russia than over either Finland or Canada, consistent with the product intercomparison from Mudryk et al. (2015).

Crocus had the smallest bias over both Canada (-22 mm) and Russia (-2.3 mm); Crocus and ERA5 had the strongest correlations over Canada (~0.7). ERA5 had the lowest RMSE and strongest correlation over Finland (33 mm, 0.8; tied with ERA-Interim/Land) and Russia (38 mm, 0.8), and the lowest bias over Finland (0.8 mm). Performance of the standalone
passive microwave products (AMSR-E) is noticeably weaker for all regions and validation statistics (with the exception of bias over Russia). RMSE for the standalone passive microwave products is nearly double that of the best performing product for both Finland and Canada, with slightly better results over Russia. For Finland and Russia, bias ranged between ±15 mm for all datasets except ERA-Interim/Land (> +20 mm) and the standalone passive microwave products and MERRA2 over Russia (+17 mm). Over Canada, bias ranged from -23 to -51 mm for all but the AMSR-E products (bias of -78 to -90 mm).
For all regions, correlation coefficients for all but the standalone passive microwave products were ~0.5 and greater. The

AMSR-E products exhibited lower or even negative correlations with snow course measurements for all three reference datasets.

We find that among GlobSnow, Crocus, ERA-Interim/Land, ERA5, GLDAS-2, MERRA and MERRA2 no individual product consistently performs best with respect to the RMSE, bias, and correlation statistics *across all regions*. This is an important finding, as it shows no clear advantage to using a single type of snow analysis, whether it is remote sensing combined with surface observations, an external snow model driven by reanalysis meteorology, or the land surface schemes within reanalyses. With higher RMSEs, greater bias, and weaker correlations relative to the other seven datasets, this assessment shows the standalone passive microwave algorithms do not perform in a comparable fashion to the other products.

To determine the influence of SWE magnitude on product performance, all three reference snow course datasets were binned into 10 mm increments for comparison with the gridded SWE estimates (Sect. 2.3, Fig. 4). Crocus and MERRA perform similarly, with reasonable agreement up to about 150 mm of SWE and a tendency to underestimate SWE for deeper snow and overestimate SWE for shallow snow. MERRA2 behaves in a similar fashion, but slightly overestimates SWE below ~150 mm. The performance of GLDAS-2 and ERA5 is similar to Crocus and MERRA except that they both underestimate SWE across a larger range of reference values (>100 mm), consistent with the negative bias in Fig. 3b (with the exception of ERA5 over Finland). GlobSnow overestimates SWE up to ~100 mm and underestimates above ~130 mm while ERA-Interim/Land overestimates SWE up to ~180 mm, consistent with the positive bias over Russia and Finland (Fig. 3b) The AMSR-E v1.0 product exhibits low sensitivity to SWE, especially for values >70 mm and overestimates low SWE values. Better results were found for the newer AMSR-E v2.0 product, although the retrievals plateau at about 100 mm, and show no sensitivity to further SWE increases.

To quantify the influence of seasonality on product performance, validation statistics (RMSE, bias, correlation) were computed at a bi-weekly time step (10 days for Russia) for 2002 through 2010 (Sect. 2.3). Figure 5 shows the monthly evolution from November through April over Russia and provides insight into both the seasonal evolution of product-specific uncertainty and the spread in uncertainty between products. In general, RMSE and bias magnitude both increase over the course of the snow season. Early in the snow season, the RMSE and bias magnitudes are low because snow is shallow, although even small errors can produce high relative RMSE. As SWE increases through the snow accumulation season, the RMSE and the spread in RMSE between products increases. While not true for every product, bias also tends to become increasingly negative over the course of the snow season. By the end of the snow season, inter-product spread in RMSE and bias are at a maximum. Peak uncertainty late in the season is driven by cumulative errors over the entire season, differences in the timing of snow melt onset, and different melt rates. Whereas the RMSE and bias evolve over the course of the snow season, the magnitude of correlation for all but the AMSR-E products is stable. This is an encouraging result as it indicates that SWE anomalies should be reasonably realistic throughout the season, even if climatological amounts of SWE differ strongly between analyses. A similar seasonal evolution of product specific uncertainties is observed for both Finland and Canada (not shown).

The analyses summarized in Figures 2–4 indicate that Crocus, MERRA and ERA5 perform slightly better than the other reanalysis-based products and GlobSnow, while the two AMSR-E products perform substantially worse. To what extent do these conclusions suggest that one should choose a single gridded SWE product as the 'best' dataset? We address this question by analyzing how the error statistics (RMSE and correlation) of multiple-product combinations compare to those of individual products. Such multi-product SWE ensembles have previously been employed to charaterize uncertainty (e.g. Mudryk et al., 2015, 2017, 2018a; Krinner et al., 2018). Here we demonstrate that such ensembles also tend to improve overall accuracy. The two AMSR-E products were excluded from this analysis because of the low correlation with snow course measurements as illustrated in Figs. 3c, 4h, 4i, and 5c. Further, for this analysis, we did not separate error statistics by country (Russia, Finland and Canada are considered on aggreate). Figure 6a confirms the conclusion that Crocus, MERRA and ERA5 perform slightly better than the other products since the average of all product combinations that involve those particular snow analyses have lower RMSE and higher correlation than averages involving the remaining products. However, we find that combinations of products often have a lower RMSE and higher correlation than individual products. For example, *any* possible combination of two or more products has improved RMSE and correlation compared to GLDAS-2, GlobSnow or ERA-Interim/Land considered individually (not shown explicitly). For MERRA and MERRA2, more than 90% of all possible combinations of two or more products have improved RMSE and correlation compared to the single product. For Crocus, approximately 40% of product combinations have improved RMSE and correlation [than the single product] while for ERA5, 70% of all possible product combinations have lower RMSE and 35% have higher correlation. This tendency for multi-product combinations to have improved accuracy is demonstrated generally in Figure 6b. As the number of products included in a multi-product combination increases, the correlation improves and the RMSE decreases with the lowest RMSE and highest correlation attained when all seven products are combined. This improvement in accuracy suggests that, to some extent, each product has randomized errors which are averaged out by considering multiple products. Because the RMSE of even the best performing products is at the margins of acceptable uncertainty for operational (<15%; Rott et al., 2010; Larue et al. 2017) and scientific (10-25%; Derksen and Nagler, 2019) requirements, the increase in accuracy represents a simple method to yield performance gains.

## 3.3 Correlation Analysis

To determine the strength of agreement among datasets, temporal and spatial correlation analysis was performed as described in Section 2.3. In preparing the datasets for intercomparison, a very strong negative trend since 1980 was found for ERA5 snow mass. This is driven by a stepwise discontinuity introduced by the assimilation of satellite derived binary snow/no-snow estimates starting in 2004 (Patricia de Rosnay, personal communication; Fig. 7). While this change addressed a positive snow extent bias during the melt season (e.g. Orsolini et al., 2019), it renders the raw ERA5 snow mass time series unsuitable for climate analysis. We therefore considered ERA5 separately from the other snow analyses (Crocus, GLDAS-2, MERRA2, ERA-Interim/Land) for the intercomparison analysis. Results obtained substituting MERRA with MERRA2 were similar so

only those including MERRA2 are presented. In the subsequent analysis R4 refers to a suite of four products (Crocus, GLDAS-2, MERRA2 and ERA-Interim/Land) that rely on reanalysis in some way.

Each of the products in the R4 suite exhibit moderately strong spatial and temporal correlations with each other (Fig. 8). The correlations, ranging between 0.5 and 0.7, represent the average of the six pairwise combinations of these four products. The agreement among these four datasets is consistent with the expected coherence of their forcing meteorologies and the relative

influence of land model and meteorological forcing on hemispheric scale snow mass previously established by Mudryk et al. (2015). While Figure 8 illustrates that the spatial patterns of ERA5 snow mass anomalies are comparable to those of GlobSnow and the R4 products, the stepwise discontinuity in its climatology lowers the correlation of its snow mass time series. It is possible to correct for this discontinuity in an *ad hoc* manner by adjusting the snow mass starting in the fall of 2004 by the difference in the climatology before and after the discontinuity. Applying this correction yields correlation values more in line

with those seen among the R4 products and GlobSnow (dashed symbol in Fig. 8). For the snow analyses and GlobSnow, the mean pattern correlation is lower than the corresponding temporal correlation of total snow mass (Fig. 8). This may be due to the presence of opposite-signed spatial biases that cancel when spatially aggregated into a snow mass time series. In contrast to the snow analyses and Globsnow, there is a lack of temporal and spatial correlation between the AMSR-E products and the R4 datasets. Spatially, this is an expected result given the differences in climatological SWE patterns shown in Fig. 1. The

weak temporal correlation means the snow mass anomalies do not evolve in-phase with the other products as the snow season evolves.

Further insight is gained through the calculation of correlation maps among groups of datasets (Fig. 9), where temporal correlations of daily SWE are calculated analogous to Northern Hemisphere snow mass but for each grid cell. As expected, the reanalysis datasets are strongly correlated to each other (R4-R4 and R4-E5 in Fig. 9). Correlations between GlobSnow and

365 the R4 products are strong across most snow covered regions of the Northern Hemisphere (GS-R4), with the exception of parts of Arctic Canada and the ephemeral snow zones of both North America and Eurasia (note that alpine areas are masked in the GlobSnow product). As noted earlier, the performance of GlobSnow is closely tied to the density of snow depth observations used as inputs to the retrievals (Larue et al., 2017; Brown et al., 2018) which likely contributes to the low correlations in parts of Arctic Canada where there are relatively few observations. The NASA AMSR-E v1.0 dataset exhibits very weak anomaly

correlations with the R4 datasets (N1-R4), and even negative correlations over the boreal forest of North America and parts of central and eastern Siberia. The AMSR-E v2.0 algorithm shows improved anomaly correlations over eastern Siberia (N2-R4; likely by better accounting for the combination of shallow snow and large snow grains found in this region (Tedesco and Jeyaratnam, 2016)) and the boreal forest of North America, although correlations remain weak over the remainder of the snow-covered northern hemisphere.

## 4. Conclusions and Discussion

In this study, we compared three types of northern hemisphere gridded SWE products: (1) those utilizing some form of reanalysis (Crocus, ERA-Interim/Land, ERA5, GLDAS-2, MERRA, MERRA2); (2) passive microwave remote sensing combined with surface observations (GlobSnow v2.0); and (3) standalone passive microwave retrievals (AMSR-E v1.0 and v2.0). There is past evidence of acceptable algorithm performance for standalone passive microwave products, particularly in open environments with relatively shallow snow (Derksen et al., 2004; Vuyovich et al., 2014), or when SWE retrievals are converted to snow cover extent (Brown et al., 2010). At the continental scale, however, the standalone AMSR-E SWE products have stark differences in climatological SWE patterns compared to other available products (see Fig. 1).

Evaluation against snow course measurements from Russia, Finland, and Canada show higher RMSE and bias, and lower correlation for standalone passive microwave products compared to the seven other datasets (Fig. 3). While uncertainty for all products tends to increase with deeper snow, this is a critical issue for the AMSR-E products because of pronounced negative bias even at relatively low SWE values (<100 mm; Fig. 4 and 5). Although there is no single product that consistently performs best over all regions with respect to bias, RMSE and correlation, Crocus and ERA5 do perform best across the range of snow conditions captured by the validation dataset. However, while a particular product may outperform others over some regions, this is no guarantee that it will do so everywhere, so we are not recommending any one product. Furthermore, we have demonstrated that averaging multiple products together tends to lead to additional accuracy improvements (Fig. 6), while as exemplified by ERA5, a single product may have properties which lend itself to one type of analysis but make it unsuitable for others.

Correlation analysis performed with respect to both space and time shows consistent behaviour with strong statistical agreement among the six reanalysis-based products and GlobSnow (consistent with Mudryk et al., 2015), which clearly benefits from the ingestion of daily surface snow depth data into the retrievals compared to the standalone passive microwave datasets. ERA5 also assimilates point snow depth observations into a state-of-the-art assimilation system and yields excellent validation results. The slightly stronger validation for ERA5 compared to GlobSnow suggests the impact of the ERA5 assimilation system, which ingests multiple data streams, improves the SWE estimates more than the impact of passive microwave remote sensing on the GlobSnow retrievals (which also assimilates point snow depth observations). However, it is important to highlight that the validation results do not convey that the raw ERA5 snow mass time series contains a significant discontinuity in 2004, caused by an abrupt change to assimilate satellite derived snow extent information. So, while ERA5 may provide one of the better SWE estimates for instantaneous applications like numerical weather prediction, the data are unsuitable (at least in an uncorrected form) for climate analysis.

As with any continental-scale evaluation, our results may (or may not) apply to small regions or local domains, and the validation results do not apply to alpine areas which contribute a large proportion (~30%; Wrzesien et al., 2019) to the total

northern hemispheric SWE. In areas of complex terrain, uncertainty in meteorological forcing within reanalyses, particularly precipitation amount and phase (Lundquist et al., 2019) must also be considered. Further, in alpine regions the coarse resolution of the gridded SWE products (25 km or more) does not lend itself to comparison with snow course observations because of limited representativeness of surface observations in complex terrain and across elevation gradients; a different validation approach is likely needed for mountain areas.

The AMSR-E products exhibit weak spatial agreement and negative temporal anomaly correlations with the other datasets (Fig. 8 and 9). The retrieval of SWE solely from passive microwave measurements is a difficult challenge, and despite the best efforts of many research groups over many decades, passive microwave-based standalone algorithms do not perform as well as other methods that make use of ancillary snow depth measurements or snow models. Although there are many attractive attributes (wide swath, all-weather imaging, long legacy time series, and theoretical sensitivity to SWE under simplified assumptions) passive microwave data has always been a measurement of opportunity for snow applications, not an ideal measurement system. This introduces intrinsic biases and errors into the standalone retrieval scheme because of the non 'optimal' nature of these measurements for snow applications.

Despite these challenges, there are opportunities to utilize satellite passive microwave measurement as a component of SWE product development moving forward. Machine learning operators show potential for the radiance-based assimilation of brightness temperatures (e.g. Forman and Reichle 2014) analogous to how L-band brightness temperatures are assimilated for improved soil moisture analyses. Assimilation approaches also show potential for addressing challenges posed by stratigraphy (Durand et al., 2011; Andreadis and Lettenmaier, 2012) and deep snow (Li et al., 2012). While coarse resolution is an inherent challenge with satellite passive microwave measurements, enhanced resolution products spanning multiple decades are now available (Long and Brodzik, 2016; Takala et al., 2017).

The combination of brightness temperature measurements, surface snow depth observations, and forward radiometric modeling are able to produce skillful SWE products. This approach was already used successfully within the ESA GlobSnow project, and will be further enhanced within the ESA Climate Change Initiative (CCI) snow project. It is important to note that the brightness temperature component of the GlobSnow/Snow CCI retrieval has direct heritage to standalone passive microwave retrieval approaches which date back to the first generation of passive microwave imagers launched in the 1970s. This also suggests that research focusing on passive microwave interactions with snow parameters should not be neglected as, ultimately, better understanding of the underlying physics is a positive step for algorithm improvement.

While the continued development of a remote sensing capabilities for SWE represents an important observational capability, it is necessary to also appreciate the quality of the large scale model derived SWE products. The combination of reanalysis meteorology and snow models yields very useful snow information, which can be refined as forcing data (particularly

precipitation) and snow models continue to improve. Only through combined and integrated improvements in remote sensing, modeling, and observations will real progress in SWE product development be achieved and sustained.

**Data availability.** Vincent Vionnet and Bertrand Decharme both provided data from the Crocus snowpack model; the NASA
AMSR-E SWE v1.0 dataset and is available from the paper's authors upon request as is the NASA AMSR-E SWE v2.0 dataset. The remaining datasets are available for download via the links and references provided in Sect. 2.

**Competing interests.** The authors declare that they have no conflicts of interest.

**Acknowledgements.** This work was conducted as a part of the European Space Agency funded Satellite Snow Product Intercomparison Exercise (SnowPEx). We appreciate the contributions from data providers: Météo-France (Crocus), NASA
Goddard Earth Sciences Data and Information Services Center (MERRA, MERRA2, GLDAS-2), European Centre for Mid-range Weather Forecasts (ERA-Interim/Land, ERA5), and the Finnish Meteorological Institute (GlobSnow). Snow course data were made available by RusHydroMet, the Finnish Environment Institute (SYKE), and the Meteorological Service of Canada. Mike Brady (ECCC) provided technical support and assistance. Thanks to the late Dr. Andrew Slater for inspiration.

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

**Table 1.** Summary of SWE products evaluated in this study.

| Data Product | Method | Ancillary/ Forcing Data | Resolution | Reference/ Availability |
|---|---|---|---|---|
| GlobSnow v2.0 | Passive microwave + in situ | Weather station snow depth measurements | 25 km | Takala et al., 2011 www.globsnow.info |
| NASA AMSR-E v1.0 | Standalone passive microwave | | 25 km | Kelly (2009) nsidc.org |
| NASA AMSR-E v2.0 | Microwave + ground station climatology | Weather station snow depth climatology | 25 km | Tedesco and Jeyaratnam (2016) nsidc.org† |
| ERA-Interim/Land | HTESSEL land surface model | ERA-interim | 0.75° x 0.75° | Balsamo et al (2015) www.ecmwf.int |
| ERA5 | HTESSEL land surface model | ERA5 | 0.25° x 0.25° | Hersbach et al. (2019) C3S (2017) |
| MERRA | Catchment land surface model | MERRA | 0.5° x 0.67° | Rienecker et al (2011) GMAO (2017a) |
| MERRA2 | Catchment land surface model | MERRA2 | 0.5° x 0.625° | Gelaro et al. (2017) GMAO (2017b) |
| Crocus | ISBA land surface + Crocus snow model | ERA-interim | 1° x 1° | Brun et al (2013) ‡ |
| GLDAS-2 | Noah 3.3 land surface model | Princeton Met. | 1° x 1° | Rodell et al (2004) disc.gsfc.nasa.gov |

† The v2 product is not available via NSIDC over the 2002–2010 period, however data using the same algorithm is available from July 2012–present. Contact authors for availability.
‡ Contact authors for availability.

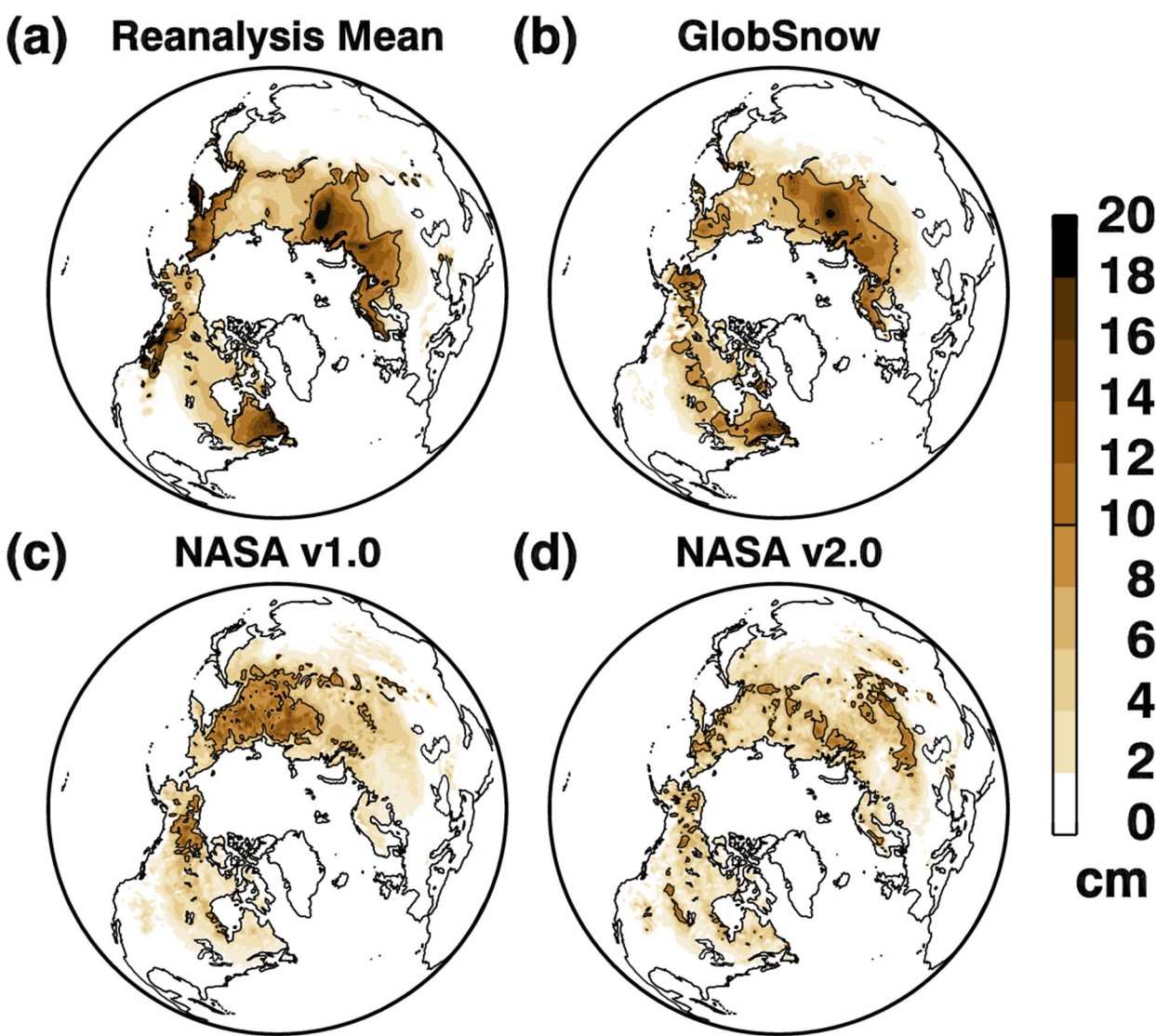

**Figure 1.** Mean January, February, and March (JFM) SWE over the 2003 – 2010 period for (a) four reanalysis driven products (GLDAS-2, ERA-Interim/Land, Crocus, and MERRA2); (b) GlobSnow v2.0; (c) NASA AMSR-E SWE v1.0; (d) NASA AMSR-E SWE v2.0.

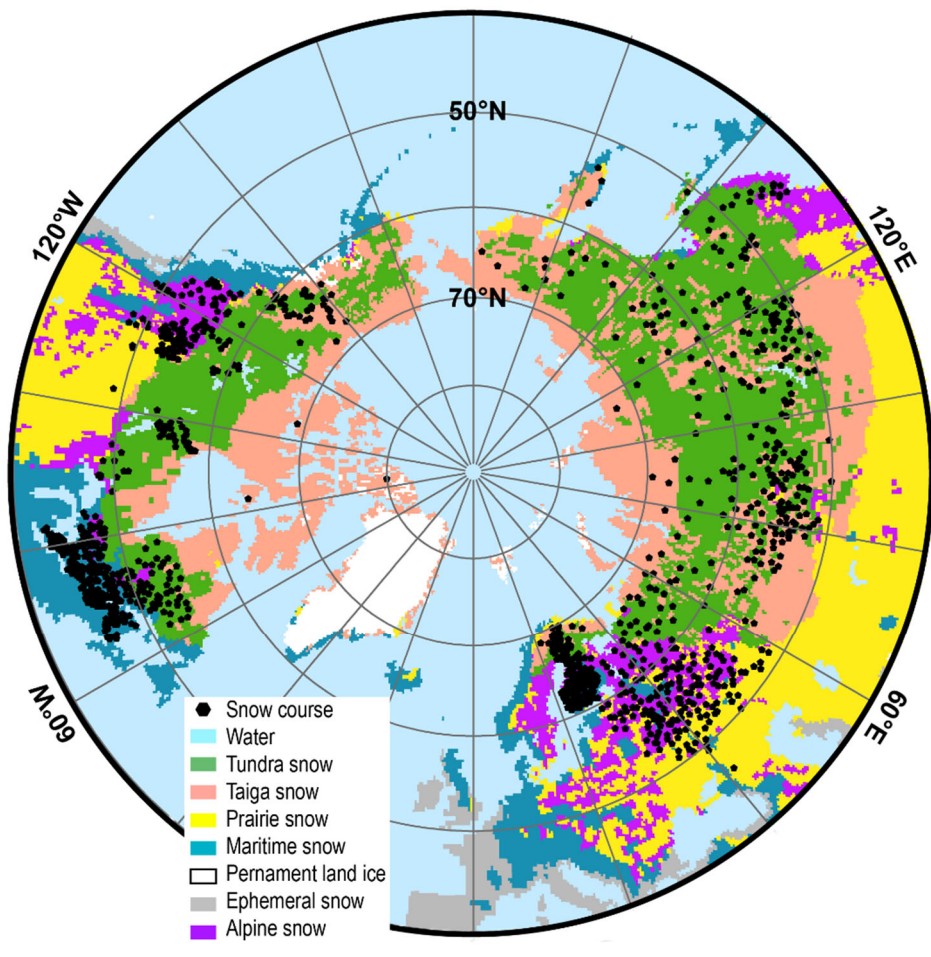

**Figure 2.** Centroid of 25 km EASE grid cells with snow course observations used in the analysis (Sect. 2.3) overlaid on snow-
climate classes (Sturm et al., 2009).

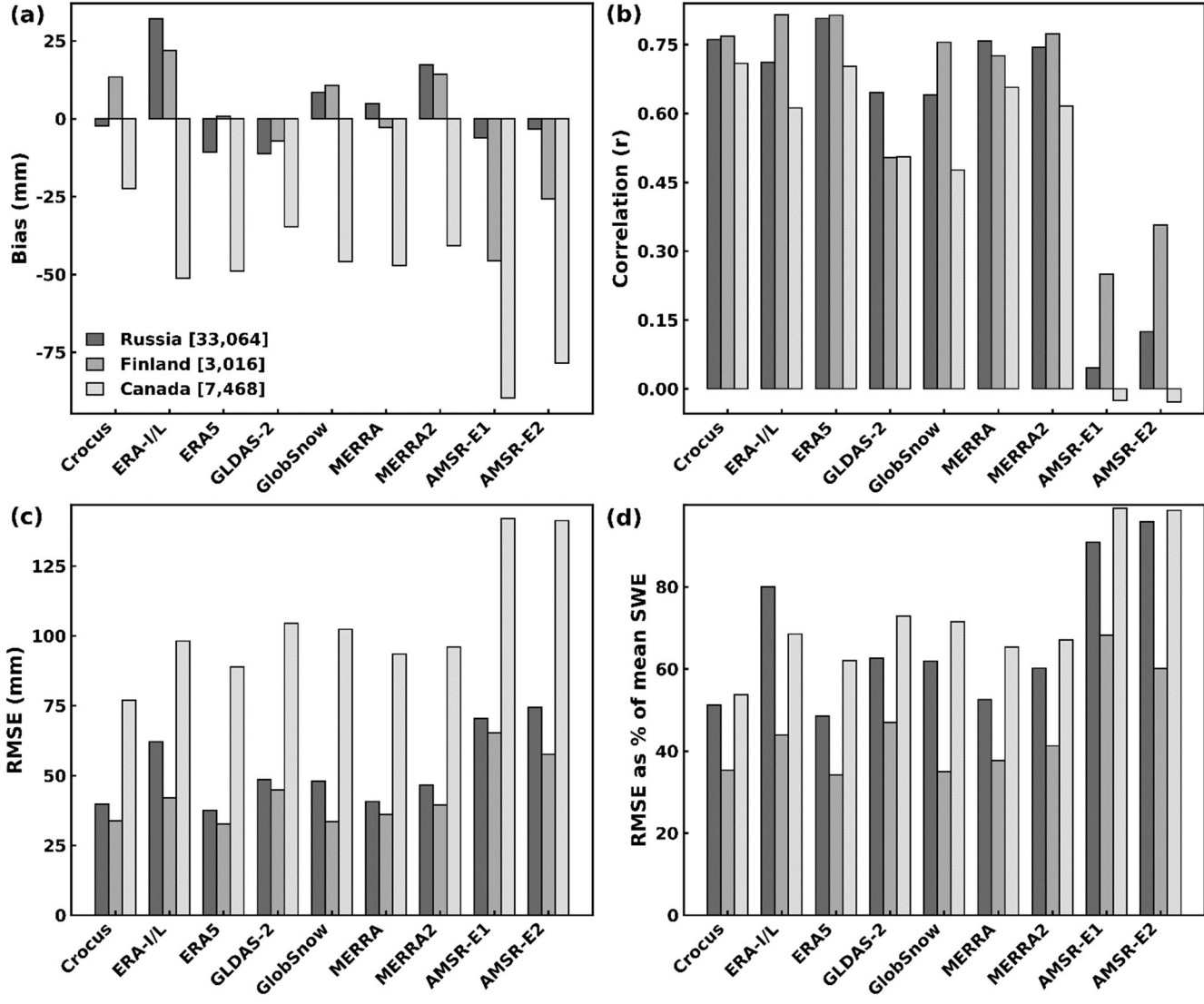

**Figure 3.** Validation statistics (a: bias; b: correlation; c: RMSE; d: RMSE as percentage of mean SWE) for the nine SWE products for November through April, 2002–2010 [ERA-I/L = ERA-Interim/Land; GlobSnow = GlobSnow v2.0]. Total number of grid cells with snow course measurements in square brackets in panel a.

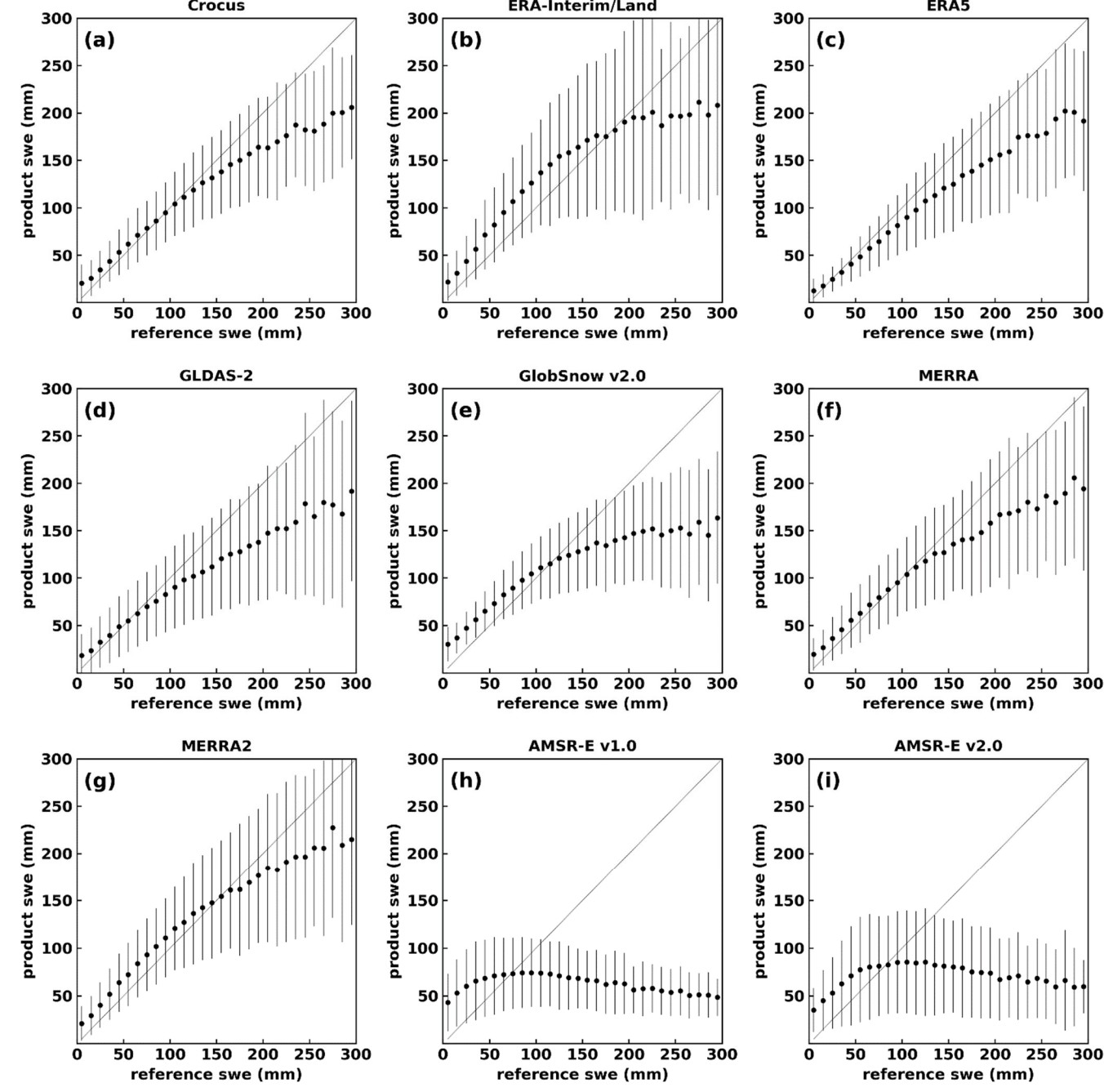

**Figure 4.** Performance of SWE datasets versus reference snow course SWE ± 1 standard deviation for (a) Crocus; (b) ERA-Interim/Land; (c) ERA5; (d) GLDAS-2; (e) GlobSnow v2.0; (f) MERRA; (g) MERRA2; (h) AMSR-E v1.0; (i) AMSR-E v2.0. SWE values above 300 mm are not shown.

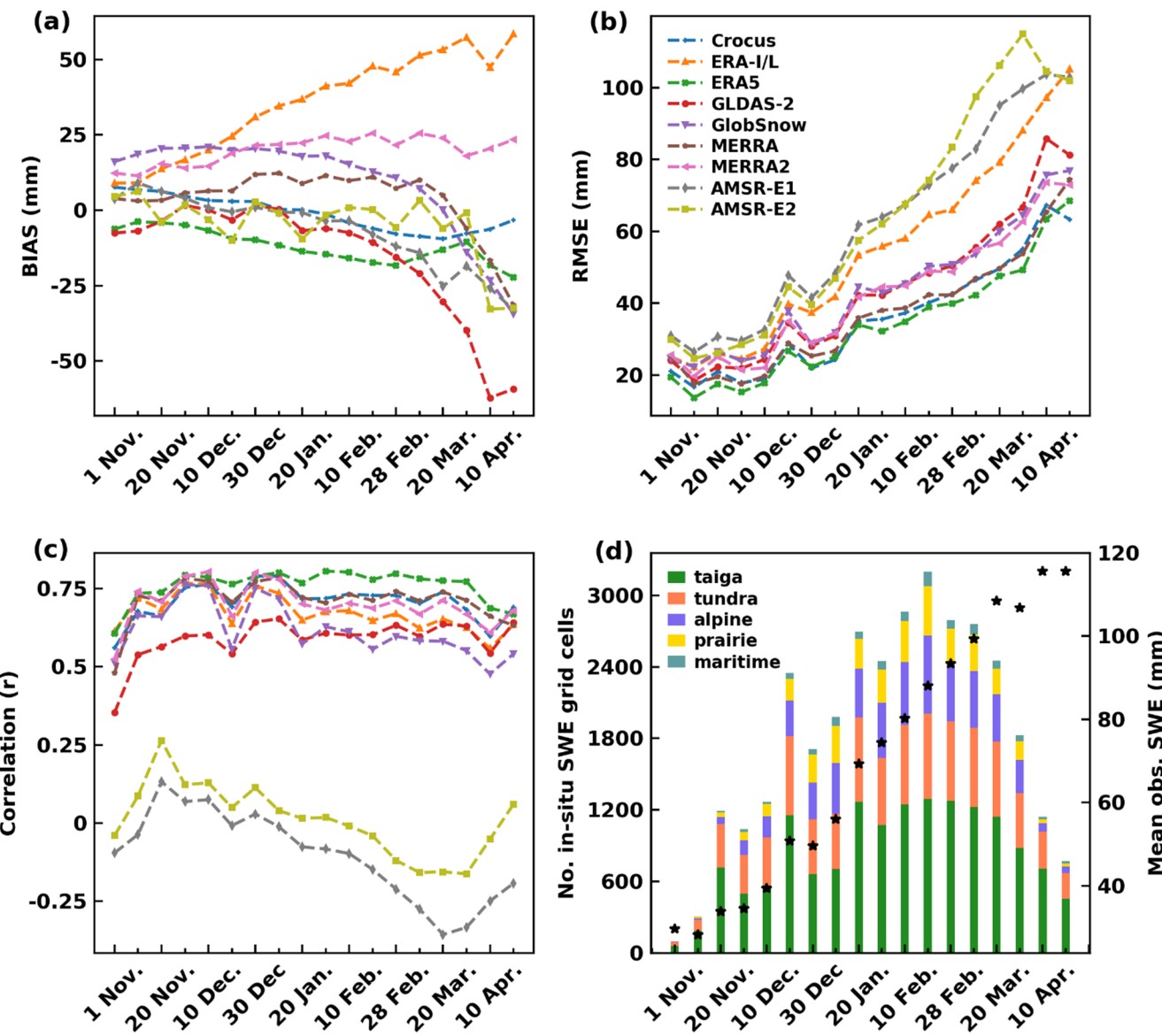

**Figure 5.** (a) Bias (b) RMSE and (c) correlation coefficient relative to the Russia snow course dataset (Sect. 2.1) for each ten day time step over the 2002–2010 period. (d) Number of grid cells with snow course observations by Sturm et al. (2009) snow class (bars, left-hand axis), mean observed SWE (stars, right-hand axis) [ERA-I/L = ERA-Interim/Land, GlobSnow = GlobSnow v2.0].

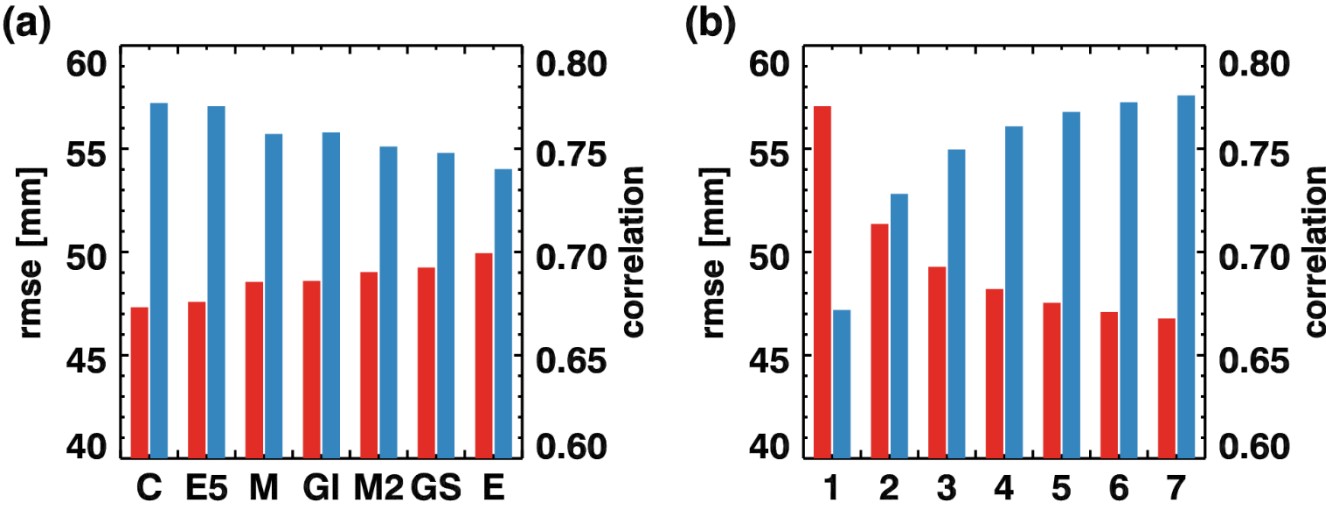

Figure 6. RMSE (red) and correlation (blue) of snow course measurements with various combinations of SWE products. (a) Average of all combinations that contain the specified individual product [C = Crocus, E5 = ERA5, M = MERRA, Gl = GLDAS-2, M2 = MERRA2, GS = GlobSnow v2.0, E = ERA-Interim/Land] and (b) average of all combinations of N products as specified on x-axis.

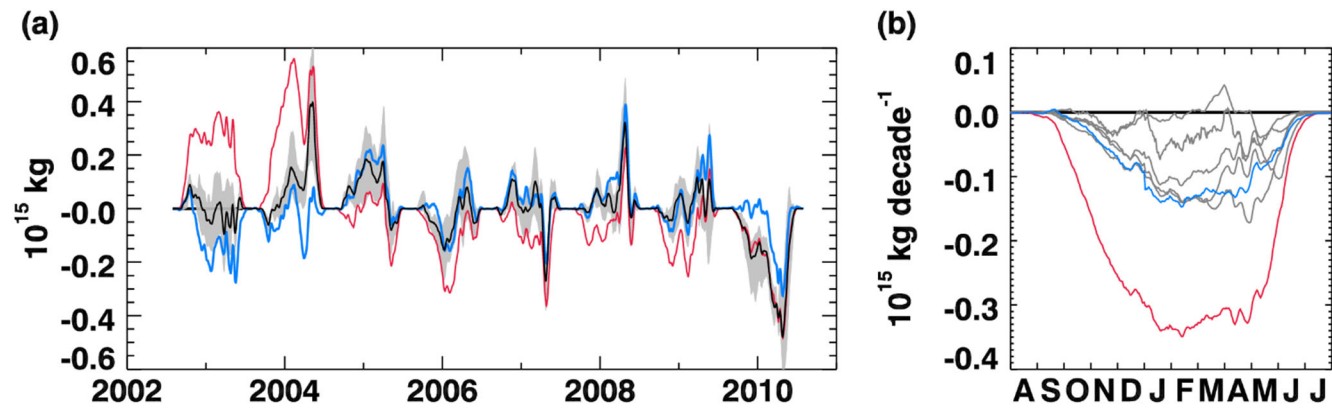

**Figure 7.** (a) Average Northern Hemisphere snow mass anomalies (black) and spread (shading) calculated from five component products: MERRA2, Crocus, GlobSnow v2.0, GLDAS-2 and ERA-Interim/Land along with snow mass anomalies from raw (red) and corrected (blue) ERA5 values. (b) Trends (1981–2010) from the five component time series used for the average in panel a (grey) along with trends from the raw (red) and corrected (blue) ERA5 time series. The ERA5 discontinuity occurs in January 2004.

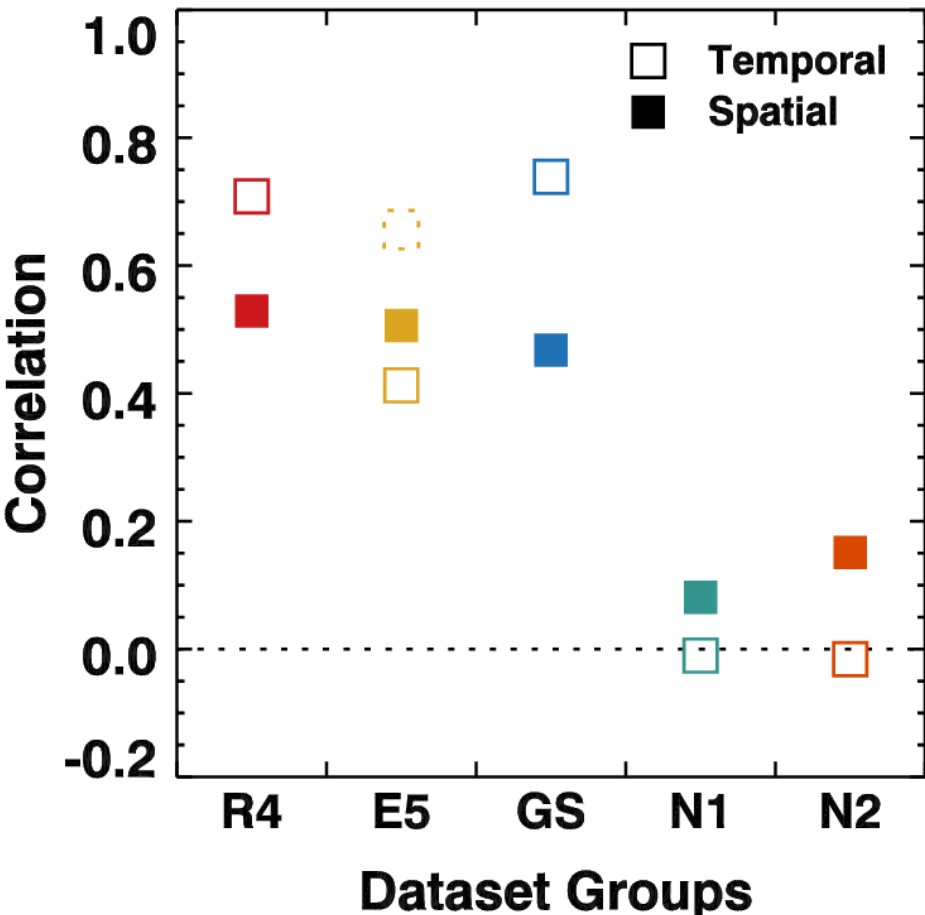

**Figure 8**. Temporal and spatial correlations among groups of products over the 2002–2010 time period. Temporal correlations assess the extent to which anomalous northern hemispheric snow mass jointly evolves between pairs of datasets while spatial correlations assess the pattern correlation of SWE fields for pairs of datasets; see text (Sect. 2.3) for details. R4 = the average of six pairwise correlations between Crocus, GLDAS-2, ERA-Interim/Land, and MERRA2. E5 = the average of 4 pairwise correlations between ERA5 and each R4 product. GS = the average of 4 pairwise correlations between GlobSnow v2.0 and each R4 product. N1 = the average of four pairwise correlations between AMSR-E v1.0 and each R4 product. N2 = the average of four pairwise correlations between AMSR-E v2.0 and each R4 product. The dotted square shows the impact of correcting the E5 snow mass anomalies for a discontinuity introduced in 2004.

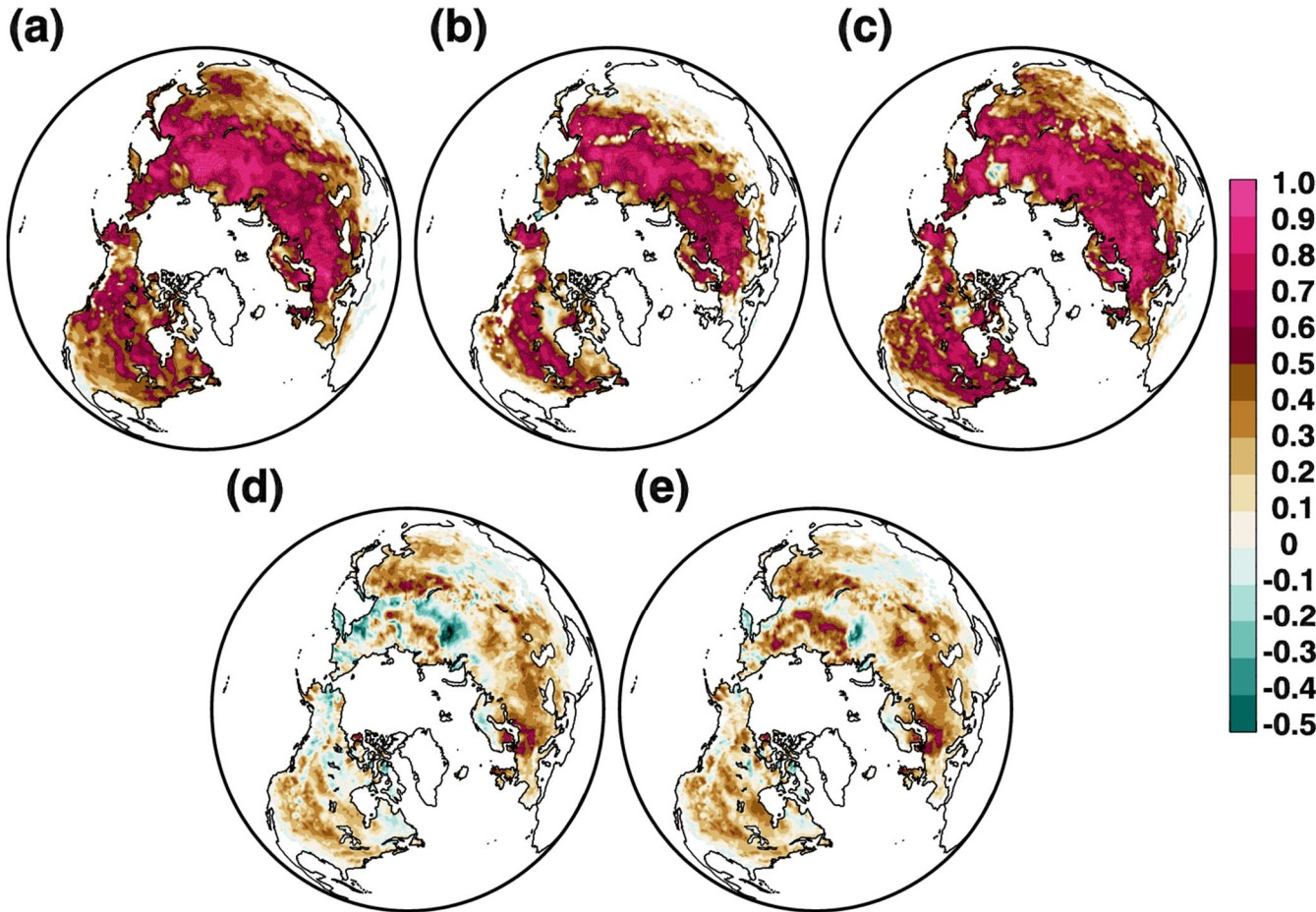

**Figure 9**. Correlation maps (2002–2010) for four reanalysis driven products (Crocus, GLDAS-2, ERA-Interim/Land, and MERRA2) relative to: (a) each other (mean correlation between the four reanalysis driven products); (b) GlobSnow v2.0; (c) ERA5; (d) NASA AMSR-E SWE v1.0; (e) NASA AMSR-E SWE v2.0.