# Peer review of "Evaluation of long term Northern Hemisphere snow water equivalent products"

_The Cryosphere, 2019_

## Referee Comment (RC1) · Anonymous Referee #1 · 3 Jan 2020

The authors discuss the evaluation of three types of Northern Hemisphere snow water equivalent (SWE) products, including (i) four reanalysis-based products, (ii) two stand-alone passive microwave remote sensing products, and (iii) one product based on a combinition of passive microwave remote sensing data and in situ snow depth measurements.

The evaluation is primarily vs. a large number of independent snow course measurements from Russia, Finland, and Canada. The authors find that the performance of the stand-alone passive microwave remote sensing products is considerably worse than that of the other products, and only the passive microwave product constrained with surface observations provides comparable performance to the reanalsysis-based products.

[Figure]

Among the reanalysis-based SWE products, MERRA and the Crocus/ERA-Interim product perform best, suggesting that these products should be included in any multi-product ensemble estimate.

The manuscript discusses an important and still active field of cryospheric research. The manuscript is not ground-breaking and hews closely to the datasets evaluated in Mudryk et al. (2015). However, it includes the AMSR-E stand-alone passive microwave remote sensing products and, if I am not mistaken, the performance evaluation vs. the snow course measurements. These new elements provide, in my opinion, sufficient novelty to warrant eventual publication of the paper in The Cryosphere. However, before I can recomment publication, the authors would need to address the MAJOR issues outlined in the comments below.

Major comments: ————

1) Dataset selection and period

a) Why evaluate MERRA data when MERRA-2 has now been available for 3+ years (Gelaro et al. 2017), and MERRA has been discontinued since early 2016??? There are some differences MERRA and MERRA-2 SWE (e.g., Reichle et al. 2017). As it stands, the reader has to assume that MERRA was used because that is the dataset that was ready to use from the earlier Mudryk et al. (2015) publication. At the very least, the authors need to discuss the existence of MERRA-2, point to the relevant literature and differences, and justify their use of MERRA instead of MERRA-2.

Gelaro et al. (2017), The Modern-Era Retrospective Analysis for Research and Applications, Version-2 (MERRA-2), Journal of Climate, 30, 5419-5454, doi:10.1175/JCLI-D-16-0758.1.

Reichle et al. (2017), Assessment of MERRA-2 land surface hydrology estimates, Journal of Climate, 30, 2937-2960, doi:10.1175/JCLI-D-16-0720.1.

A similar comment applies to the paper's use of ERA-Land and "Crocus", which are

both based on ERA-Interim, which has been replaced by ERA-5 and ERA-5/Land (albeit much more recently than the MERRA version change).

b) Why does the analysis stop in 2010 (Table 1)? As far as I am aware, all of the SWE products should be available for several years beyond 2010 (given that AMSR2 extends the AMSR-E record to the present, with only a modest gap). Being a few years behind real-time was ok in Mudryk et al. (2015), but by now 2010 nearly a decade behind real-time, which at the very least requires justification.

2) The discussion of the methodology needs to be improved.

a) As it stands, there are bits and pieces of the methodology in the Results section, and the Methods section is lacking a concise discussion of the various metrics. E.g., lines 193-196, 235-236, and 276-282 belong in the Methods section, and the Methods section needs a complete discussion of the metrics.

b) The temporal and spatial resolution of the metrics calculations is a bit unclear. Line 128 states that all SWE products were regridded onto a 1-deg grid, whereas the snow course measurements are on the 25-km EASE grid (line 161). How are the 1-deg grid cells matched with the 25-km EASE grid cells? And why introduce the 25-km EASE grid in the first place, given that the snow course data are not anywhere near that scale (transects range from 150 m to 4 km), and in any case the 25-km EASE grid is different from the 1-deg grid of the SWE products. Why not use the same grid for the SWE products and the (gridded) transect data? At the very least, this requires justification and clarification.

c) The snow course data are available from once every 5 days to once every month (Section 2.1), whereas the SWE products are available between hourly and daily (which requires better clarification!). Lines 158-161 state that the snow course observations were "converted into bi-weekly [or (over Russia) ten-day] periods". How exactly are the SWE products and snow course data matched in time for the computation of the metrics? Are the SWE products sampled on a single day (1st and 15th

of each month), or are two-week (or ten-day) average SWE values computed from the hourly/daily products before the metrics are computed? This needs to be clarified.

d) Lines 138-141: Please clarify whether snow course data are measurements of snow depth or SWE. (The paragraph in question talks a lot about snow depth, but only in the context of the point-scale measurements used in GlobSnow.) Also, if snow course data are snow *depth* measurements, how are the measurements converted to SWE? Using local and contemporaneous snow density measurements? Or climatological snow density values?

e) Fig 2c: It is not clear how the correlation shown here was computed. Is this the spatial average of the temporal correlation coefficient at the individual grid cells? Or the spatial correlation of the time series average? Or all data points thrown into a single correlation coefficient calculation???

f) Lines 235-236: "seasonality [metrics...] were computed at a bi-weekly time step for 2002 through 2010". This is unclear. Based on this statement, the metrics could have been computed in one of the following ways: - subset time series at each location, then throw all values into the metrics computation - subset time series at each location, then compute (temporal) metrics at each location, then spatially average metrics - subset time series at each location, then compute time-average SWE values, then compute (spatial) metrics Which is it?

g) How were zero SWE values treated? Are SWE values excluded from the metrics computations if the snow course and/or SWE product indicated zero SWE? How about cross-masking

h) The number of grid cells ("locations") with snow course data is unclear. According to section 2.1, there are 517 snow course locations in Russia, 200 in Finland, and >1000 in Canada. However, the y-axis scale in Fig. 4d suggests that at most ∼100 locations are used for Russia. The discrepancy between 517 and ∼100 needs to be discussed explicitly. Is this reduction due to insufficient length of time series, or because the snow

course data are ultimately averaged into 25-km EASE grid cells (or 1-deg grid cells)???
How many sites (or grid cells?) were used for Finland and Canada?

i) Lines 276-279: The text here is unclear. Is the metric discussed in line 277 different
from that discussed in line 278? In Line 281, in which way are the "anomolous SWE
fields" different from the "anomalous snow mass"??? Is not "SWE" synonymous with
"snow mass"?? (Or does "snow mass" here refer to the spatially integrated SWE? If
so, that is not clear.)

Also, throughout section 3.3 I was confused whether there were two different temporal
correlation metrics (one using raw data including the seasonal cycle, and another using
data with the mean seasonal cycle removed).

3) ERA-Land and Crocus similarities, and dependence on snow measurements

a) ERA-Land and Crocus use the same forcing data. Including the correlation of the
two datasets in Fig 6 therefore artificially elevates the "R4" correlation. Should the
ERA-Land/Crocus pair not be excluded from the correlation coefficients contributing to
the "R4" value?

b) Perhaps more importantly, ERA-Land and Crocus are *not* fully independent of in
situ snow measurements. Both datasets rely on ERA-Interim surface meteorological
forcing data. ERA-Interim includes a snow analysis that is based on snow cover data
and on in situ snow depth measurements, which impacts the ERA-Interim surface me-
teorology estimates through, at the least, surface albedo feedback. This needs to be
pointed out. (Note that there is no snow analysis in MERRA or MERRA-2.)

4) Lines 75-76 (implicitly) motivates the present study by saying that "[t]o date, these
ensembles have relied heavily on models driven by atmospheric analysis and include
only a single dataset (GlobSnow) which utilizes remote sensing." However, Line 263
states that "[t]he two AMSR-E products were excluded from this comparison because
of the low correlation with the snow course data [...]" That is, the present study is not

really different from previous studies in this regard. This particular motivation of the present study seems therefore invalid.

Minor comments: ————

i) Line 52: Please add a reference for the "temporal inconsistencies" in reanalysis datasets, e.g., Robertson, F. R., M. G. Bosilovich, J. Chen, and T. L. Miller, 2011: The effect of satellite observing system changes on MERRA water and energy fluxes. J. Climate, 24, 5197–5217, doi:10.1175/2011JCLI4227.1

ii) Lines 53-61: Recent results using Sentinel-1 (active) radar data suggest that at least for deep mountain snow much higher spatial resolution snow depth estimates are achievable (Lievens et al. 2019). This should at least be pointed out here, and a clarification should be added that the present study focuses on passive microwave data only. The Lievens et al. (2019) results also suggest that the text in Line 77 may need clarification. Lievens et al. (2019), Snow depth variability in the Northern Hemisphere mountains observed from space, Nature Communications, 10, 4629, doi:10.1038/s41467-019-12566-y.

iii) The nomenclature "NASA Historical" and "NASA Operational" is a bit unfortunate. First, MERRA is (or rather, was) also a *NASA* (quasi-)operational product. Second, the use of *Historical* and *Operational* suggests that "Historical" is only for the retrospective period while "Operational" is for the present and future. However, if I understand the manuscript correctly, "Historical" is really an older version of the NASA AMSR-E retrieval product, and "Operational" is a newer version of that same product. Two of the authors of the present paper are also authors of the "NASA AMSR-E" product. They should know the appropriate version numbers of the NASA AMSR-E products discussed here, and these version numbers should be used in the paper.

iv) In the context of Figure 2 or the corresponding Methods discussion, the number of grid cells with snow course measurements contributing to the metrics should be provided. See also comment 2h) above.

v) Line 346: replace "idealized" with "ideal"

vi) Lines 369-370: The term "NASA AMSR-E *operational* dataset" appears twice, once in each line. Should one of the two be the "historical" dataset?

vii) Line 82: replace "to evaluation" with "to evaluate"

viii) Lines 123-124: The paper should make it clear whether the SWE output from the reanalysis data was used or whether the snow depth output was used (with subsequent conversion to SWE using ancillary snow density values). This is a bit unclear.

---

## Referee Comment (RC2) · Anonymous Referee #2 · 12 Jan 2020

Summary This manuscript performs an intercomparisons and evaluation of seven different northern hemisphere representations of daily Snow Water Equivalent: four reanalyses (CROCUS, ERA/Land, MERRA, and GLDAS, two products based on AMSR-E passive microwave data, and the GlobSnow product, which is based on a combination of passive microwave and in situ snow data. The authors compare the products to one another and find broad similarities among all products except the passive microwave only products, which are quite different. Evaluation against in situ snow course data also suggest that all products other than the passive microwave only datasets provide similar levels of accuracy. The study has implications for any hemisphere-scale analysis that relies on understanding of snowpack.

Overall Review

I found this paper to quite well written, and I very much like the approach the authors took to their analysis. I kept finding myself wanting a particular type of analysis to be done and then, a few paragraphs later, the authors had done just what I'd hoped for (e.g. the ensemble analysis). However, there are a few points that I think would substantially improve the paper.

The first one, and most major, is that the selection of reanalysis products is somewhat outdated. In particular, both MERRA and ERA-Interim/Land have been, at least to some degree, superseded by MERRA-2 and both ERA5 and ERA5-Land. In the latter case, the resolution of the data products is higher (30 km) and much higher (9 km). I anticipate that most users in the future will probably use these more recent datasets rather than the older ones listed here. So the current paper is useful, but it would be so much more useful if these additional datasets were included. I recognize that it would probably be a fair amount of work to add them in, but I really think it would probably be worth it. That said, this is a decision that should be made by the authors in consultation with the editor—I do think the paper is publishable as is, just not as useful as it could be.

Second, I would like to see just a bit more discussion of snow in high-topography regions. I recognize that this is not the primary focus of the paper, but mountain snowpack is pretty important. There's been some really good work published on this recently—I'm thinking of the paper by Jessica Lundquist that talks about the utility of models vs. observations in understanding mountain snow and precipitation (https://doi.org/10.1175/BAMS-D-19-0001.1) and some of the work by Melissa Wrzesien that intercompares different global products in a way similar to what's done here (but explicitly for mountains), such as https://doi.org/10.1029/2019WR025350. I don't think this needs to be a very heavy lift, but I would like to see some mention in the abstract of the fact that mountains are (mostly) excluded in the analysis, along with a paragraph in the discussion addressing this point and related work.

Specific comments

Line 34: "There is a growing number"

Line 39: I think it would be good to cite the relevant paper by Meromy et al. (2013) here: https://doi.org/10.1002/hyp.9355

Line 60: I just want to say that I really like this sentence about gridded/in situ datset comparisons

Line 76: Somewhere in here it would probably be good to mention the new Nature Communications paper by Lievens et al. (https://doi.org/10.1038/s41467-019-12566-y). Also would be good to mention it in the section on mountains that I suggest above.

Line 160: what fraction of the grid cells have at least one data point? How do these data represent (vs. not represent) different environments?

Line 164 (Section 3.1): It would be great if you could get a little bit more quantitative in this section. Right now it seems like you're doing a visual comparison of the climatologies from the different datasets, but it wouldn't be difficult to also compare them quantitatively.

Line 181: "The source of inability of the standalone passive microwave products" sounds a bit awkward. What about "The reason the standalone passive microwave products. . . " or something similar?

Line 185: There should be a hyphen between observation and sparse.

Line 203: I think it might make sense to include a metric such as relative RMSE or normalized RMSE to assess whether the performance in Canada is, in fact worse because there's more snow. I also wonder if it might not have something to do with the less systematic nature of the in situ measurements in Canada. You even make reference to relative RMSE later in the paper (Line 239), though no values are provided.

Line 279: I had to read this sentence a bunch of times before I understood what you meant. Could you rewrite to try to be a bit clearer about what you did? I think you

basically took the spatial correlations for all days and then averaged them. Also, could you clarify what difference, if any, there is between spatial correlations and pattern correlations? I think you're using them interchangeably, but it's not totally clear.

Line 283: If you look at the AMSR-E datasets in Figure 6, it sure looks like the mean pattern correlation is higher. Can you clarify?

Line 333: No need for the comma between Canada and show.

---

## Referee Comment (RC3) · Anonymous Referee #3 · 20 Jan 2020

General comments

The authors conducted a thorough analysis of various publicly-available SWE products. Continental SWE datasets based on reanalysis products, land surface modeling, and passive microwave satellite data are evaluated against (in-situ transect) snow course measurements. The standalone passive microwave-based datasets seem to perform poorly relative to the snow course measurements and the other SWE products. Although no 'best' dataset is identified, the product ensembles that contained Crocus or MERRA performed better.

The authors have provided a sufficiently explanatory literature review for readers who may be unfamiliar with SWE estimation techniques. This effort will benefit researchers who utilize SWE products as ancillary information in their models and help them un-

derstand the uncertainty associated with these products.

The paper is well-written and requires only minor modifications.

Specific comments

a) It would be helpful to add a figure showing the geolocation of the snow course data in Section 2.1. It will add topological context to the analysis.

b) Please add quantitative and/or qualitative information regarding the uncertainty or tentative precision of the snow course measurements used for evaluation in Section 2.1. A quantitative measure of uncertainty for each of the three datasets would be sufficiently descriptive.

c) Increasing the size of individual stereographic maps will improve the visual clarity of the figures.

d) Please highlight in the conclusion (Line 341) that this analysis is for continental performance evaluation and may/may not apply to small regional or local domains.

Technical corrections

Abstract – A '0.1' increase in correlation does not seem very significant. Please provide justification of the significance of this increase in correlation using additional analysis (such as hypothesis testing). This analysis can be included as a separate paragraph in Section 3.3. If no justification exists, then it would be advisable to remove the sentence from the abstract.

Line 32 – Please define which seasonal forecasts the authors are eluding to. Being specific will make the discussion more accessible to the reader.

Line 38 – Please define the difference between 'snow depth' and 'surface snowfall' measurements.

Line 40 – It isn't clear what 'coarse' grid cells means. Can you quantify what you mean
by 'coarse'?

Line 83 – Replace evaluation with evaluate.

Line 110 – Rephrase the sentence to highlight the difference between using separate brightness channels versus spectral difference for SWE estimation.

Line 162 – Please add an appropriate reference for the EASE2 grid.

Line 220 – Please elaborate briefly on what the 'acceptable' uncertainty is for SWE. Please be specific in terms of quantitative (rather than qualitative) values of SWE uncertainty.

Line 225 – GlobSnow also underestimates SWE above values > ∼130mm. This statement needs to be included here.

Line 239 – Figure 3g shows >70mm rather than >60mm as the pivot point.

Line 266 – contain is written twice.

Figure – 1: Extra 'E' in AMSR-E in caption.

Figure – 3: a) Please define how Figure 3 was developed in the main text. Is the binning based on average SWE values for each grid cell or the average of bi-weekly values for all snow courses? b) The term 'retrieval' is used in the caption. This term does not apply to all the different SWE datasets being evaluated. Please change the sentence from 'retrieval performance versus reference SWE ....' to 'Performance of SWE datasets versus reference SWE measurements .....' c) The figure labels for subplots f and g do not match the caption labels.

Figure – 4: In the text, a bi-weekly time step is specified while the caption describes a ten day time step. Please clarify this contradiction.

Figure – 6: The figure title can be removed since the caption and labels are self-explanatory.

Data availability There seems to be a typing error in the data availability description. The NASA AMSR-E operational dataset is mentioned twice.

---

## Editor Comment (EC1) · Florent Dominé (Editor) · 3 Feb 2020

Dear Authors,

The reviewers have made many positive comments about your paper. However, most reviewers have noted that more recent products, such as MERRA-2 and ERA5, are available and would have been more appropriate for your analysis. Another issue is that your analysis stops in 2010. It seems that the ideal response to these comments would be to present updated analyses using more recent products and to extend your work to a more recent date. I realize that a much easier option for you would be to formulate some justification for your current approach. I am sure you realize that the impact and interest of your paper would be considerably enhanced if you did chose to

go the hardest route. Furthermore, The Cryosphere is also much more interested in more up to date studies. Thank you for considering these aspects in your response to comments.

Best regards

Florent Domine

---

## Author Comment (AC1) · 28 Feb 2020

**Response to Editor**

Thanks to all three reviewers for their constructive comments on the manuscript. Our revisions have resulted in the inclusion of additional datasets, and improved the clarity and impact of the paper.

As outlined in more detail in our responses to the review comments, we have incorporated both ERA5 and MERRA2 for the validation (comparison with snow course measurements). We retained both MERRA and ERA-land in order to show the difference in performance between subsequent generations.

For the dataset inter-comparison, we have replaced MERRA with MERRA2 and added ERA5. Because ERA5 represents a significant departure in many ways from ERA-land (optimal interpolation vs. Cressman, a number of bug fixes, data assimilation of IMS etc.) we have retained ERA-land in this analysis. In preparing ERA5 data for the inter-comparison analysis, we found a very strong negative trend since 1980, which is driven by a discontinuity in the time series starting in 2004. After following up with ECMWF, it is clear that this is caused by the assimilation of IMS snow extent data into ERA5, which starts in 2004. This means ERA5 cannot be used (without some form of correction) for snow mass trend analysis. This is an important finding to communicate to the snow community, which we now emphasize in the revised manuscript.

Our response to each comment is outlined below in **bold**. Revised text is in *red italics*. We hope these responses are clear, and we look forward to submitting the revised manuscript.

**Anonymous Referee #1**

**Received and published: 3 January 2020**

The authors discuss the evaluation of three types of Northern Hemisphere snow water equivalent (SWE) products, including (i) four reanalysis-based products, (ii) two stand-alone passive microwave remote sensing products, and (iii) one product based on a combination of passive microwave remote sensing data and in situ snow depth measurements.

The evaluation is primarily vs. a large number of independent snow course measurements from Russia, Finland, and Canada. The authors find that the performance of the stand-alone passive microwave remote sensing products is considerably worse than that of the other products, and only the passive microwave product constrained with surface observations provides comparable performance to the reanalysis-based products.

Among the reanalysis-based SWE products, MERRA and the Crocus/ERA-Interim product perform best, suggesting that these products should be included in any multi-product ensemble estimate.

The manuscript discusses an important and still active field of cryospheric research. The manuscript is not ground-breaking and hews closely to the datasets evaluated in Mudryk et al. (2015). However, it includes the AMSR-E stand-alone passive microwave remote sensing products and, if I am not mistaken, the performance evaluation vs. the snow course measurements. These new elements provide, in my opinion, sufficient novelty to warrant eventual publication of the paper in The Cryosphere. However, before I can recommend publication, the authors would need to address the MAJOR issues outlined in the comments below.

**Major comments:**

**1) Dataset selection and period**

a) Why evaluate MERRA data when MERRA-2 has now been available for 3+ years (Gelaro et al. 2017), and MERRA has been discontinued since early 2016??? There are some differences MERRA and MERRA-2 SWE (e.g., Reichle et al. 2017). As it stands, the reader has to assume that MERRA was used because that is the dataset that was ready to use from the earlier Mudryk et al. (2015) publication. At the very least, the authors need to discuss the existence of MERRA-2, point to the relevant literature and differences, and justify their use of MERRA instead of MERRA-2.

Gelaro et al. (2017), The Modern-Era Retrospective Analysis for Research and Applications, Version-2 (MERRA-2), Journal of Climate, 30, 5419-5454, doi:10.1175/JCLI-D-16-0758.1.

Reichle et al. (2017), Assessment of MERRA-2 land surface hydrology estimates, Journal of Climate, 30, 2937-2960, doi:10.1175/JCLI-D-16-0720.1.

This is an important comment, and this issue was raised by other reviewers as well. We have updated the analysis to now include both MERRA2 and ERA5; the manuscript was revised in many places (including Table 1 and all figures) to reflect this new analysis. The new versions of the figures are included at the end of this document. We retained both MERRA and ERA-land in the validation in order to show the difference in performance between subsequent generations of the same product. In the case of ERA5, there is noticeable improvement, especially across Eurasia where the positive SWE bias in ERA-land is corrected in ERA5. This difference is likely

due in large part to the assimilation of weather station snow depth observations in ERA5, so text was added to emphasize the impact of this change. Changes from MERRA to MERRA2 are much more subtle, and based on the validation statistics it appears snow mass in MERRA2 is actually degraded slightly from MERRA.

For the dataset inter-comparison, we have replaced MERRA with MERRA2 and added ERA5 since it represents a significant departure in many ways to ERA-land. In preparing ERA5 data for the inter-comparison analysis, we found a very strong negative trend since 1980, which is driven by a discontinuity in the time series starting in 2004. After following up with ECMWF, it is clear that this is caused by the assimilation of IMS snow extent data into ERA5, which starts in 2004. This means ERA5 cannot be used (without some form of correction) for trend analysis. This is an important finding to communicate to the snow community, which we now include in the revised manuscript.

A similar comment applies to the paper's use of ERA-Land and "Crocus", which are both based on ERA-Interim, which has been replaced by ERA-5 and ERA-5/Land (albeit much more recently than the MERRA version change).

See our response above. We have revised the analysis to include MERRA2 and ERA5. Manuscript and figures have been revised to reflect this change.

While ERA5 is now available and could be used with Crocus, the most recent version of the Crocus dataset is still forced with ERA-interim. This will be changed in future versions of this product, but not until an evaluation is completed at Météo France on the impact of changes in the forcing dataset on the SWE simulations. The important attribute of the Crocus dataset is that it includes a more complex physical snow model compared to the other products. In that sense the forcing dataset is of secondary importance, so we have retained the Crocus product as part of the analysis.

b) Why does the analysis stop in 2010 (Table 1)? As far as I am aware, all of the SWE products should be available for several years beyond 2010 (given that AMSR2 extends the AMSR-E record to the present, with only a modest gap). Being a few years behind real-time was ok in Mudryk et al. (2015), but by now 2010 nearly a decade behind real-time, which at the very least requires justification.

The SWE products are not all available over a common time period. The 2002-2010 period was used to maximize commonality between datasets. The primary time series limitations are GLDAS-2 and ERA-Interim/Land (which both end in 2010) and v1 of the AMSR-E product (which only covers 2002-2011). While it is desirable in some ways to cover the most recent time period possible, the focus of this analysis is on the validation and inter-comparison of the products. In that sense, only a sufficiently long time period is required (in order to capture the range of naturally varying snow conditions) but the actual time period covered is less important. Text added (Line 101-102):

'The analyses described subsequently in Section 2.3 were conducted for the period 2002-2010 to maximize commonality between products.'

2) The discussion of the methodology needs to be improved.

a) As it stands, there are bits and pieces of the methodology in the Results section, and the Methods section is lacking a concise discussion of the various metrics. E.g., lines 193-196, 235-236, and 276-282 belong in the Methods section, and the Methods section needs a complete discussion of the metrics.

**We have revised Section 2 to now cover both datasets and methods:**

Section 2. Datasets and methods Section 2.1. Gridded SWE products Section 2.2. Snow course data Section 2.3 Validation and inter-comparison methods

This change included moving text that was previously in the Results (lines 193-196; 235-236, and 276-282) into Section 2. Additional clarification to the methods text in Section 2.3 was also added in response to other points mentioned in the review comments.

- In Section 2.1, which describes the gridded SWE products, we added discussion of how snow depth observations are used in different reanalysis products.
- Section 2.2 provides a more comprehensive discussion of the snow course data than was previously included. We improved our explanation of snow course measurement protocol, added mention of snow course measurement uncertainty, and explicitly stated how we dealt with zero SWE values.
- Section 2.3 clearly outlines the two approaches we have taken to evaluating SWE products validation and inter-comparison. Much of this text was either previously in the results section or is new in response to reviewer feedback. We use the term *validation* to represent the evaluation of gridded products against snow course data as a measure of ground truth; whereas *inter-comparison* is similar to the analysis of Mudryk et al. (2015) and quantifies the spatial and temporal anomaly correlations between datasets. Methods, including quantitative metrics, relevant to each of the two approaches are outlined separately.

b) The **temporal and spatial resolution of the metrics calculations is a bit unclear.** Line128 states that all SWE products were regridded onto a 1-deg grid, whereas the snow course measurements are on the 25-km EASE grid (line 161). How are the 1-deg grid cells matched with the 25-km EASE grid cells?

The re-gridded (1°x1°) products were only used for inter-comparison. Native product resolutions were used for validation with the gridded snow course data. We have included more precise wording and additional clarification in the revised methods section – see response above. Relevant sections:

**Line 200-203:** 'For a given measurement date, each EASE grid cell with available snow course data was paired with corresponding SWE values from each of the nine gridded products. The paired SWE values correspond to the grid cell at each product's native resolution that intersects with the centroid of the EASE grid cell snow course measurement.'

**Line 214-216:** 'The intercomparison analysis does not consider the snow course measurements, only the nine gridded products. For this analysis daily frequency SWE from each product was interpolated to a regular 1° x 1° longitude–latitude grid.'

And why introduce the 25-km EASE grid in the first place, given that the snow course data are not anywhere near that scale(transects range from 150 m to 4 km), and in any case the 25-km EASE grid is

different from the 1-deg grid of the SWE products. Why not use the same grid for the SWE products and the (gridded) transect data? At the very least, this requires justification and clarification.

We gridded the snow course measurements in order to reduce sampling bias due to concentrations of surveys in some areas (this is a particularly important issue for the Canadian data because snow courses are concentrated in heavily populated regions of southern Canada). We chose the 25 km EASE-Grid as a compromise resolution which reduced spatial sampling bias but didn't introduce too much uncertainty due to the representativeness of snow course measurements relative to the coarse grid cell resolution. We have clarified this in the revised methods section lines 206-224.

'For the validation analysis, SWE product grid cells must be matched in both space and time with the snow course measurements. To achieve this, snow course observations from Canada and Finland were first grouped into bi-weekly periods using a 16 day window centred on the 1st or 15th of each month. Likewise, over Russia, observations were grouped into ten-day periods centred on the typical measurement dates (10th, 20th, 30th of each month). For each temporal grouping, snow course measurements falling within a given 25 x 25 km EASE grid cell (Brodzik et al. 2012) were averaged together, thereby forming a gridded snow course field. (Fig. 2) The majority (69%) of EASE grid cells used in our analysis had only one snow course observation, 25% had two snow course observation and the remaining 5% had three or more snow course observations. Grouping the snow course data had the largest impact over Canada and Russia where 35% and 20% of grid cells, respectively, had more than one snow course observation. Finland's snow course network is representative of the landscape's different snow environments, defined by Sturm snow classes, In Canada, and to a lesser extent over Russia, tundra environments which are often remote, are under-sampled while maritime and alpine snow types are oversampled.

For a given measurement date, each EASE grid cell with available snow course data was paired with corresponding SWE values from each of the nine gridded products. The paired SWE values correspond to the grid cell at each product's native resolution that intersects with the centroid of the EASE grid cell snow course measurement. In order to fairly compare how the gridded products perform against one another, only snow course data from EASE grid cells where there exist corresponding paired values from all nine of the SWE products were analyzed. For example, regions of complex topography are implicitly excluded from the validation analysis because they are masked in GlobSnow.'

c) The snow course data are available from once every 5 days to once every month (Section 2.1), whereas the SWE products are available between hourly and daily (which requires better clarification!). Lines 158-161 state that the snow course observations were "converted into bi-weekly [or (over Russia) ten-day] periods". How exactly are the SWE products and snow course data matched in time for the computation of the metrics? Are the SWE products sampled on a single day (1st and 15th of each month), or are two-week (or ten-day) average SWE values computed from the hourly/daily products before the metrics are computed? This needs to be clarified.

Part 1: The snow course data are available from once every 5 days to once every month (Section 2.1), whereas the SWE products are available between hourly and daily (which requires better clarification!).

We have added text describing how we used data from products with sub-daily values. Lines 102-106:

'All the products provide SWE directly, and are available at daily or sub-daily frequency. For the four products available at sub-daily frequency, we either obtained daily mean versions directly from the product's distribution sites (MERRA, MERRA-2) or sampled a single sub-daily snapshot for each calendar day ERA-Interim/Land, ERA-5) which we consider to be representative of the daily mean value.'

Part 2: How exactly are the SWE products and snow course data matched in time for the computation of the metrics? Are the SWE products sampled on a single day (1st and 15th of each month), or are two-week (or ten-day) average SWE values computed from the hourly/daily products before the metrics are computed? This needs to be clarified.

Snow course measurements are compared with a single day from the SWE products. For example, SWE products from 1 February are compared with the gridded snow course data centered on 1 February. We have added additional clarification concerning the matching of SWE products and gridded in situ data and calculation of validation metrics:

**Lines 207-217:** 'For the validation analysis, SWE product grid cells must be matched in both space and time with the snow course measurements. To achieve this, snow course observations from Canada and Finland were first grouped into bi-weekly periods using a 16 day window centred on the 1st or 15th of each month. Likewise, over Russia, observations were grouped into ten-day periods centred on the typical measurement dates (10th, 20th, 30th of each month). For each temporal grouping, snow course measurements falling within a given 25 x 25 km EASE grid cell (Brodzik et al. 2012) were averaged together, thereby forming a gridded snow course field.'

**Lines 218-224:** 'For a given measurement date, each EASE grid cell with available snow course data was paired with corresponding SWE values from each of the nine gridded products. The paired SWE values correspond to the grid cell at each product's native resolution that intersects with the centroid of the EASE grid cell snow course measurement. In order to fairly compare how the gridded products perform against one another, only snow course data from EASE grid cells where there exist corresponding paired values from all nine of the SWE products were analyzed. For example, regions of complex topography are implicitly excluded from the validation analysis because they are masked in GlobSnow.'

d) Lines 138-141: Please clarify whether snow course data are measurements of snow depth or SWE. (The paragraph in question talks a lot about snow depth, but only in the context of the point-scale measurements used in GlobSnow.) Also, if snow course data are snow \*depth\* measurements, how are the measurements converted to SWE? Using local and contemporaneous snow density measurements? Or climatological snow density values?

Thank you for this comment as our description was not clear. Snow course measurements are not snow depth measurements but composed of a series of manual gravimetric snow corer measurements. The precise instrument and protocol differs by jurisdiction but a snow cylinder (snow corer) is used to collect a vertically integrated sample of the snowpack at a specific location that is then weighed. For a given snow course, the reported SWE is the average of the multiple gravimetric measurements (specific averaging protocol may vary (e.g. Brown et al., 2019; Haberkorn 2019). In contrast, the point snow depth measurements used in GlobSnow are single depth measurements from networks of climate/weather stations and do not contain any information about snow density nor SWE. These point snow depth values are assimilated directly into some products (GlobSnow; ERA5) but are fully independent from the manual snow course measurements used for validation.

Brown, R., and Braaten, R.: Spatial and temporal variability of Canadian monthly snow depths, 1946-1995, Atmos.-Ocean, 36, 37–54. https://doi.org/10.1080/07055900.1998.9649605, 1998.

Haberkorn, A. (Ed.): *European Snow Booklet*, 363 pp., doi:10.16904/envidat.59, 2019.

We have revised the text (Lines 158-166) to better differentiate between the snow course SWE measurements and point snow depth measurements:

'The key results in this paper stem from evaluation of the suite of products described in Section 2.1 with a network of in situ snow course measurements from multiple national and sub-national agencies. These data consist of manual gravimetric snow measurements made at multiple locations along a pre-defined transect that are averaged to obtain a single SWE value for a given snow course on a given day. Measurements are collected along the same transect multiple times each snow season. By averaging multiple samples along a transect, the resulting SWE measurement provides better representation of subgrid scale variability than a single point measurement and is thereby more suitable for evaluation of SWE at the scale of the gridded products. These snow course data are fully independent of the point snow depth measurements assimilated into GlobSnow and ERA5. Transect length, number of samples collected along each transect, and sample aggregation methods differ among reporting agencies as described below.'

e) Fig 2c: It is not clear how the correlation shown here was computed. Is this the spatial average of the temporal correlation coefficient at the individual grid cells? Or the spatial correlation of the time series average? Or all data points thrown into a single correlation coefficient calculation???

It is the latter. Correlation coefficient (r) is calculated from all data pairs. Line 177-179 in methods was added for clarification.

**Line 224-225:** 'Bias and root mean squared error (RMSE) were calculated for each product-snow course pair and then averaged over the full 2002–2010 time period; correlation was calculated from all data pairs for the 2002–2010 period.'

f) Lines 235-236: "seasonality [metrics...] were computed at a bi-weekly time step for 2002 through 2010". This is unclear. Based on this statement, the metrics could have been computed in one of the following ways: - subset time series at each location, then throw all values into the metrics computation - subset time series at each location, then compute (temporal) metrics at each location, then spatially average metrics – subset time series at each location, then compute time-average SWE values, then compute (spatial) metrics Which is it?

Thank you for raising this. We agree this was not clear and have added clarification in Section 2.2. Metrics were computed from all data points for a given time step (1st or 15th of each month or 10, 20, 30th in the case of Russia) across all years. Revised text (Lines 226-227):

'To understand the influence of seasonality on product performance, bias, RMSE and correlation were also computed across all years for each biweekly period (10 day period for Russia).'

g) How were zero SWE values treated? Are SWE values excluded from the metrics computations if the snow course and/or SWE product indicated zero SWE? How about cross-masking

Snow course measurements are not made when there is no snow (zero snow measurements are not typically reported). To avoid the use of any potentially erroneous in situ SWE values, zero snow course measurements were removed prior to the spatiotemporal aggregation described in Section 2.3. For Nov-April 2002-2010 Russia had 17 zero SWE observations, Finland and Canada had none. Added text Lines 182-185:

'Because snow courses are only conducted during the snow season, they are not a reliable a measure of SWE absence nor are zero SWE values reported in a consistent manner across all jurisdictions. To ensure consistency, all snow course observations were removed prior to spatiotemporal aggregation. SWE product zero values were also excluded.'

h) The number of grid cells ("locations") with snow course data is unclear. According to section 2.1, there are 517 snow course locations in Russia, 200 in Finland, and >1000 in Canada. However, the y-axis scale in Fig. 4d suggests that at most~100 locations are used for Russia. The discrepancy between 517 and ~100 needs to be discussed explicitly. Is this reduction due to insufficient length of time series, or because the snow course data are ultimately averaged into 25-km EASE grid cells (or 1-deg grid cells)??? How many sites (or grid cells?) were used for Finland and Canada?

**We think there may have been some confusion about the y-axis. The number of grid cells (dashed line) is shown on the right-hand axis and ranges from ~600 for 1 November to >3000 in January-March.**

i) Lines 276-279: The text here is unclear. Is the metric discussed in line 277 different from that discussed in line 278? In Line 281, in which way are the "anomalous SWE fields" different from the "anomalous snow mass"??? Is not "SWE" synonymous with "snow mass"??? (Or does "snow mass" here refer to the spatially integrated SWE? If so, that is not clear.)

Clarification of anomalous snow mass is included in the improved methods section 2.3. Relevant lines:

**Line 235-236:** 'To determine the strength of agreement among datasets we use three metrics, all considered for anomalous SWE or snow mass (i.e. with the seasonal cycle removed).'

**Line 239-242:** 'First, we considered the correlation between each product's time series of anomalous daily Northern Hemisphere snow mass (SWE integrated over the entire Northern Hemisphere land area).'

Also, throughout section 3.3 I was confused whether there were two different temporal correlation metrics (one using raw data including the seasonal cycle, and another using data with the mean seasonal cycle removed).

The seasonal cycle is removed for all metrics used in the inter-comparison section. All metrics (temporal correlation of domain-integrated snow mass, spatial correlation over the full domain, and temporal correlation of individual grid cell SWE) use anomalous SWE or snow mass which removes the seasonal cycle. We have clarified this in our improved methods section 2.3.

**Line 235-236:** 'To determine the strength of agreement among datasets we use three metrics, all considered for anomalous SWE or snow mass (i.e. with the seasonal cycle removed).'

3) ERA-Land and Crocus similarities, and dependence on snow measurements

a) ERA-Land and Crocus use the same forcing data. Including the correlation of the two datasets in Fig 6 therefore artificially elevates the "R4" correlation. Should the ERA-Land/Crocus pair not be excluded from the correlation coefficients contributing to the "R4" value?

While ERA-land and Crocus both use the same reanalysis forcing, (ERA-interim) the objective of using both datasets is to show the impact of the different snow models used within each dataset. The land surface model in ERA-land is HTESSEL, which has a comparatively simple snow scheme compared to the Crocus snow model. Previous work (see Figure 11 in Mudryk et al., 2015) has shown that the same land surface model with different atmospheric forcing will result in different SWE. Likewise, the same forcing applied to different land surface models will result in different SWE. This latter case is readily apparently in the different validation statistics that we present for ERA-land versus Crocus. Despite using the same forcing, the resulting SWE datasets are quite different.

b) Perhaps more importantly, ERA-Land and Crocus are \*not\* fully independent of insitu snow measurements. Both datasets rely on ERA-Interim surface meteorological forcing data. ERA-Interim includes a snow analysis that is based on snow cover data and on in situ snow depth measurements, which impacts the ERA-Interim surface meteorology estimates through, at the least, surface albedo feedback. This needs to be pointed out. (Note that there is no snow analysis in MERRA or MERRA-2.)

It's an important detail that the forcing meteorology from ERA-int includes explicit assimilation of snow information (even though the SWE produced by ERA-land and Crocus do not). For these products, the use of snow depth information is one step removed from the final SWE estimates compared to products like GlobSnow and ERA5, although the assimilation of snow information can improve variables such as lower tropospheric temperatures which obviously have an impact on snow. We have added text starting on line 143 to clarify this.

'The impact of snow depth observations also differs between reanalysis products. Snow depth observations are directly assimilated into ERA5. For ERA-Interim/Land, however, only the forcing meteorology includes explicit assimilation of point snow depth measurements (the SWE produced by ERA-land does not). Therefore, for ERA-land, the use of snow depth information is one step removed from the final SWE estimates compared to ERA5, although the assimilation of snow information can improve variables such as lower tropospheric temperatures which obviously have an indirect impact on snow.'

4) Lines 75-76 (implicitly) motivates the present study by saying that "[t]o date, these ensembles have relied heavily on models driven by atmospheric analysis and include only a single dataset (GlobSnow) which utilizes remote sensing." However, Line 263states that "[t]he two AMSR-E products were excluded from this comparison because of the low correlation with the snow course data [...]" That is, the present study is not really different from previous studies in this regard. This particular motivation of the present study seems therefore invalid.

In previous SWE ensembles used for climate studies, GlobSnow was the only component dataset which included earth observation data. This is explicitly noted on lines 81-82. So, a motivation of this study is to see if additional satellite-derived datasets (either or both of the AMSR-E products) are suitable for inclusion, but they must first be assessed, as noted on lines 86-87. The results of the validation and inter-comparison in this study clearly show that they should NOT be included as part of a historical SWE ensemble. We think this motivation is appropriate as described in the last paragraph of Section 1.

**Minor comments: -----**

i) Line 52: Please add a reference for the "temporal inconsistencies" in reanalysis datasets, e.g., Robertson, F. R., M. G. Bosilovich, J. Chen, and T. L. Miller, 2011: The effect of satellite observing system changes on MERRA water and energy fluxes. J. Climate, 24, 5197–5217, doi:10.1175/2011JCLI4227.1

**Reference to Robertson et al. 2011 added to end of sentence beginning on line 52:** 'There may also be temporal inconsistencies in the forcing data related to changes in the observational streams assimilated in the reanalyses (Robertson et al., 2011).'

**Full citation has been added to the reference list:**

Robertson, F. R., Bosilovich, M. G., Chen, J., and Miller, T. L.: The effect of satellite observing system changes on MERRA water and energy fluxes, J. Clim., 24, 5197–5217, doi:10.1175/2011JCLI4227.1, 2011

ii) Lines 53-61: Recent results using Sentinel-1 (active) radar data suggest that at least for deep mountain snow much higher spatial resolution snow depth estimates are achievable (Lievens et al. 2019). This should at least be pointed out here, and a clarification should be added that the present study focuses on passive microwave data only. The Lievens et al. (2019) results also suggest that the text in Line 77may need clarification. Lievens et al. (2019), Snow depth variability in the Northern Hemisphere mountains observed from space, Nature Communications, 10, 4629,doi:10.1038/s41467-019-12566-y.

The Lievens et al. (2019) manuscript presents positive results on the potential for C-band SAR to provide high spatial resolution snow depth estimates in mountain areas. It lacks, however, any physical explanation for how this is possible using cross-polarization C-band radar data. As such, we feel these results must be approached with some caution. Since cross-pol C-band radar data are only available since the launch of Sentinel-1A in 2014, there is limited potential to provide climate-relevant time series at the present time. Despite these limitations, we agree that it is appropriate to note the potential in this area, so we have added some text and a reference to the Lievens et al paper starting on line 62.

'There may be potential for cross-polarized C-band SAR to provide high spatial resolution snow depth estimate in mountain areas (Lievens et al., 2019), but these estimates currently lack a physical explanation. Since cross-pol C-band radar data are only available since the launch of Sentinel-1A in 2014, there is limited potential to provide climate-relevant time series.'

iii) The nomenclature "NASA Historical" and "NASA Operational" is a bit unfortunate. First, MERRA is (or rather, was) also a \*NASA\* (quasi-)operational product. Second, the use of \*Historical\* and \*Operational\* suggests that "Historical" is only for the retrospective period while "Operational" is for the present and future. However, if I understand the manuscript correctly, "Historical" is really an older version of the NASAAMSR-E retrieval product, and "Operational" is a newer version of that same product. Two of the authors of the present paper are also authors of the "NASA AMSR-E" product. They should know the appropriate version numbers of the NASA AMSR-E products discussed here, and these version numbers should be used in the paper.

We have revised the nomenclature throughout. 'NASA Historical' and 'NASA Operational' are now referred to as 'NASA AMSR-E SWE v1.0' and 'NASA AMSR-E SWE v2.0', respectively. iv) In the context of **Figure 2** or the corresponding Methods discussion, the number of grid cells with snow course measurements contributing to the metrics should be provided. See also comment 2h) above.

**The number of grid cells with snow course measurements contributing to the metrics in Fig.2 has been added to a revised Figure 2, which is now Figure 3 (see appendix).**

v) Line 346: replace "idealized" with "ideal"

**Change made.**

vi) Lines 369-370: The term "NASA AMSR-E \*operational\* dataset" appears twice, once in each line. Should one of the two be the "historical" dataset?

The first instance of NASA AMSR-E operational should have read NASA AMSR-E historical. This line has been changed to use version numbers and reads (Line 482-484):

*'...the NASA AMSR-E SWE v1.0 dataset and is available from the paper's authors upon request as is the NASA AMSR-E SWE v2.0 dataset.'*

vii) Line 82: replace "to evaluation" with "to evaluate"

**Change made.**

viii) Lines 123-124: The paper should make it clear whether the SWE output from the reanalysis data was used or whether the snow depth output was used (with subsequent conversion to SWE using ancillary snow density values). This is a bit unclear.

SWE was the output variable used for analysis. This is now noted explicitly in line 232-233.

'For this analysis daily frequency SWE from each product was interpolated to a regular 1° x 1° longitude–latitude grid.'

**Anonymous Referee #2**

**Received and published: 12 January 2020**

Summary This manuscript performs an intercomparisons and evaluation of seven different northern hemisphere representations of daily Snow Water Equivalent: four reanalyses (CROCUS, ERA/Land, MERRA, and GLDAS, two products based on AMSRE passive microwave data, and the GlobSnow product, which is based on a combination of passive microwave and in situ snow data. The authors compare the products to one another and find broad similarities among all products except the passive microwave only products, which are quite different. Evaluation against in situ snow course data also suggest that all products other than the passive microwave only datasets provide similar levels of accuracy. The study has implications for any hemisphere-scale analysis that relies on understanding of snowpack.

**Overall Review**

I found this paper to quite well written, and I very much like the approach the authors took to their analysis. I kept finding myself wanting a particular type of analysis to be done and then, a few paragraphs later, the authors had done just what I'd hoped for (e.g. the ensemble analysis). However, there are a few points that I think would substantially improve the paper. The first one, and most major, is that the selection of reanalysis products is somewhat outdated. In particular, both MERRA and ERA-Interim/Land have been, at least to some degree, superseded by MERRA-2 and both ERA5 and ERA5-Land. In the latter case, the resolution of the data products is higher (30 km) and much higher (9 km). I anticipate that most users in the future will probably use these more recent datasets rather than the older ones listed here. So the current paper is useful, but it would be so much more useful if these additional datasets were included. I recognize that it would probably be a fair amount of work to add them in, but I really think it would probably be worth it. That said, this is a decision that should be made by the authors in consultation with the editor. I do think the paper is publishable as is, just not as useful as it could be.

**As discussed in more detail in our response to Reviewer 1, we have revised the analysis to also include MERRA2 and ERA5.**

Second, I would like to see just a bit more discussion of snow in high-topography regions. I recognize that this is not the primary focus of the paper, but mountain snowpack is pretty important. There's been some really good work published on this recently. I'm thinking of the paper by Jessica Lundquist that talks about the utility of models vs. observations in understanding mountain snow and precipitation (https://doi.org/10.1175/BAMS-D-19-0001.1) and some of the work by Melissa Wrzesien that intercompares different global products in a way similar to what's done here (but explicitly for mountains), such as https://doi.org/10.1029/2019WR025350. I don't think this needs to be a very heavy lift, but I would like to see some mention in the abstract of the fact that mountains are (mostly) excluded in the analysis, along with a paragraph in the discussion addressing this point and related work.

**We now more clearly state in our conclusions (lines 434-436) that non-alpine areas are the focus of this study. We have also added text with new references in Section 4 (starting on line 445) to highlight the key issues regarding mountain snow:**

-representativeness of surface observations in complex terrain

-coarse resolution of gridded SWE products

-uncertainty in meteorological forcing, particularly precipitation amount and phase

**Lines 445-452:**

'As with any continental-scale evaluation our results may (or may not) apply to small regions or local domains and the validation results do not apply to non-alpine areas which constitute a large proportion (~30%; Wrzesien et al., 2019) of the total hemispheric SWE. The coarse resolution of the products (25 km or more) combined with the validation approach (comparison with snow course observations) is particularly problematic for mountain areas. Key sources of uncertainty include the representativeness of surface observations in complex terrain, and uncertainty in meteorological forcing from reanalyses, particularly precipitation amount and phase (Lundquist et al., 2019).'

**Citations added to the reference list:**

Lundquist, J., M. Hughes, E. Gutmann, and S. Kapnick, 2019: Our Skill in Modeling Mountain Rain and Snow is Bypassing the Skill of Our Observational Networks. Bull. Amer. Meteor. Soc., 100, 2473–2490, https://doi.org/10.1175/BAMS-D-19-0001.1

Wrzesien, M. L., Pavelsky, T. M., Durand, M. T., Dozier, J., and Lundquist, J. D.: Characterizing biases in mountain snow accumulation from global data sets, Water Res. Res., 55, 9873–9891, https://doi.org/10.1029.2019WR025350, 2019.

**Specific comments**

Line 34: "There is a growing number" **Change made.**

Line 39: I think it would be good to cite the relevant paper by Meromy et al. (2013) here: https://doi.org/10.1002/hyp.9355

**Added reference to Meromy et al. (2013) to line 41 and to reference list.**

**Line 40:** 'However, both snow depth and snowfall measurements from single point locations are intrinsically limited by a lack of confidence in how they capture the landscape mean across coarse grid cells (Meromy et al., 2013).'

**New reference:**

Meromy, L., Molotch, N. P., Link, T. E., Fassnacht, S. R., and Rice R.: Subgrid variability of snow water equivalent at operational snow stations in the western USA, Hydrological Processes, 27, 2383–2400, https://doi.org/10.1002/hyp.9355, 2012

Line 60: I just want to say that I really like this sentence about gridded/in situ dataset comparisons **Thank you!**

Line 76: Somewhere in here it would probably be good to mention the new Nature Communications paper by Lievens et al. (https://doi.org/10.1038/s41467-019-12566-y). Also would be good to mention it in the section on mountains that I suggest above.

See our response to Reviewer 1. While we have reservations about the physical mechanisms driving C-band radar response to SWE, we have added the citation with some new text to the Introduction (line 62).

Line 160: what fraction of the grid cells have at least one data point? How do these data represent (vs. not represent) different environments?

**Thank you for raising this. We have addressed each question separately below.**

Part 1: what fraction of the grid cells have at least one data point?

Only a small fraction of EASE grid cells have at least one snow course observation. This is one of the primary reasons for combining the validation and inter-comparison approaches in our overall methodology. Although more than one quarter (27%) EASE grid cells in Finland have at least one snow course observation, the situation is very different over Russia and Canada which are considerably larger and have more remote areas. Excluding permanent land ice and large water bodies but including alpine areas that are masked out in GlobSnow, only ~3-4% and 1-2% of Canada and Russia's land area, respectively, has a corresponding EASE grid cell with a snow course observation.

The majority of grid cells retained for our analysis only had one snow course measurement. Slightly less than one third (30%) had two or more snow courses, in these cases the grid cell in situ SWE value was the average of multiple snow courses. The proportion of grid cells with two or more snow courses was higher for Russia (35%) and Canada (20%) compared to Finland (7%). In Canada this occurs (primarily) because there are snow courses in close proximity to one another, this is especially the case near population centres. In contrast, the large proportion of grid cells having more than one snow course observation is attributed to there being multiple snow courses over different land cover types (field, forest, gulley) that are assigned the same WMO Id and thus have the same coordinates.

We have added a brief summary of these numbers to the end of Section 2.3 (212-215).

'The majority (69%) of EASE grid cells used in our analysis had only one snow course observation, 25% had two snow course observation and the remaining 5% had three or more snow course observations. Grouping the snow course data had the largest impact over Canada and Russia where 35% and 20% of grid cells, respectively, had more than one snow course observation.'

Part 2: How do these data represent (vs. not represent) different environments?

Snow course transects are several hundred metres to several kilometers in length and are often designed to sample multiple land cover types (e.g. the Finnish snow courses). This makes it difficult to assign each snow course to a single land cover type so instead we used the Sturm snow climate classes (Sturm et al., 1995). For each jurisdiction the proportion of each snow class (excluding permanent ice cover and large water bodies) that sampled is as follows:

Table. Percent of EASE grid cells that have at least one snow course observation by Sturm snow class [percent of grid cells with snow course observations by Sturm Snow Class / percent of land area by Sturm snow class] \*\* ^

| Sturm Snow Class | Canada             | Finland             | Russia             |
|------------------|--------------------|---------------------|--------------------|
| Tundra           | 0.54 [5.59/36.17]  | 24.39 [7.25/8.03]   | 1.15 [20.82/31.80] |
| Taiga            | 2.16 [23.31/38.10] | 22.11 [15.22/18.63] | 1.38 [38.18/48.66] |
| Maritime         | 14.2 [47.01/11.69] | 30.37 [42.03/37.45] | 3.22 [4.12/2.26]   |

| Prairie | 3.27 [4.05/4.37]  | 42.11 [5.80/3.73] | 3.02 [14.53/8.50] |
|---------|-------------------|-------------------|-------------------|
| Alpine* | 8.14 [20.04/8.69] | 25 [29.7/32.16]   | 4.53 [22.34/8.70] |

\*note: alpine snow class is not the same as the topographic mountain mask used in GlobSnow so presence of this snow type is not unexpected.

\*\*land area included in the mountain mask that is excluded in our analysis is included in the above calculations but snow courses falling in these areas are excluded.

Athere are no snow courses in the ephemeral snow zone which makes up <1% of both Canada and Russia so it is not included here.

From this analysis we see that Finland's snow course network does a good job of sampling the distribution of snow types. In Canada, tundra is heavily under-sampled while maritime and alpine snow are oversampled. The same is true over Russia but the over/under-sampling is less pronounced.

We have added a brief summary of these numbers to the end of Section 2.3 (lines 215-217) and a figure with locations of grid cells having snow course measurements overlaid on Sturm Snow Classes (new Figure 2).

'Finland's snow course network is representative of the landscape's different snow environments, defined by Sturm snow classes, In Canada, and to a lesser extent over Russia, tundra environments which are often remote, are under-sampled while maritime and alpine snow types are oversampled.'

Sturm, M., Holmgren, J., and Liston, G.: A seasonal snow cover classification system for local to global applications, Journal of Climate, 8: 1261-1283, 1995.

Line 164 (Section 3.1): It would be great if you could get a little bit more quantitative in this section. Right now it seems like you're doing a visual comparison of the climatologies from the different datasets, but it wouldn't be difficult to also compare them quantitatively.

We could compare the climatologies quantitatively by producing difference plots, but we wanted to avoid adding too many panels to the figure. The purpose of Figure 1 is to frame the analysis that follows, and show that it is readily apparent visually that the AMSR-E products differ clearly from the other datasets.

Line 181: "The source of inability of the standalone passive microwave products" sounds a bit awkward. What about "The reason the standalone passive microwave products. . . " or something similar?

Thank you for this suggestion. Revised text (Line 272):

'The reason the standalone passive microwave products fail to capture higher SWE in western Siberia, Russia, northern Europe, and eastern Canada is less clear...'

Line 185: There should be a hyphen between observation and sparse. **Change made.**

Line 203: I think it might make sense to include a metric such as relative RMSE or normalized RMSE to assess whether the performance in Canada is, in fact worse because there's more snow. I also wonder if it might not have something to do with the less systematic nature of the in situ measurements in

Canada. You even make reference to relative RMSE later in the paper (Line 239), though no values are provided.

We have added a new panel to Figure 2 (now Figure 3, see appendix) showing the RMSE as a percentage of the mean observed SWE. Mean observed SWE was taken as the average observed SWE from grid cells having snow course measurements and coincident SWE product data for all nine products. There are different statistical approaches to this calculation, but we feel this most clearly illustrates how RMSE varies as a function of SWE magnitude as measured at the snow courses.

**We have clarified the text in question (Lines 296-300):**

'Larger absolute bias and RMSE over Canada may be attributed, in part, to a higher average SWE (mean of all snow course observations > 140 mm compared to

*Figure 2. Centroid of 25km EASE grid cells with snow course observations used in the analysis (Sec. 2.3) overlaid on Sturm snow class (Strum et al. 2009).*

Data source added to reference list:

Sturm, M., Holmgren, J., Liston, G.: Global Seasonal Snow Classification System. Version 1.0. UCAR/NCAR - Earth Observing Laboratory. https://doi.org/10.5065/D69G5JX5, 2009. Accessed 14 Feb 2020.

b) Please add quantitative and/or qualitative information regarding the uncertainty or tentative precision of the snow course measurements used for evaluation in Section 2.1. A quantitative measure of uncertainty for each of the three datasets would be sufficiently descriptive.

It is difficult to attach specific uncertainty values to the snow survey measurements because non-standard sampling tools are used between snow course networks (e.g. no consistent snow corer diameter). Furthermore, error in the individual measurements is no doubt overwhelmed by uncertainty in how the snow course measurements represents the landscape mean at the scale of the gridded SWE products. We have added text which identifies the sources of uncertainty into the final paragraph of Section 2.2 (Lines 185-190):

'Finally, it is difficult to attach specific uncertainty values to the snow survey measurements because non-standard sampling tools are used between snow courses (e.g. no consistent snow corer diameter). Full discussion of snow course measurement protocols and instruments is available elsewhere (Goodison et al., 1981; Brown et al., 2019; Haberkorn et al., 2019), but it is virtually certain that uncertainty associated with the individual measurements ( $\pm$  approximately 5%; Brown et al., 2019) is overwhelmed by uncertainty in how the snow course measurements represent the landscape mean at the scale of the gridded SWE products.'

**Complete citation for the following references added to reference list:**

Goodison, B. E., Ferguson, H. L., McKay, G. A.: Measurement and data analysis. In: Handbook of Snow (D. M. Gray and D. H. Male, eds.), pp. 191–274. Reprint. Caldwell, NJ, USA, The Blackburn Press, 1981.

Haberkorn, A. (Ed.): European Snow Booklet, 363 pp., doi:10.16904/envidat.59, 2019.

c) Increasing the size of individual stereographic maps will improve the visual clarity of the figures.

**We have increased the size of the stereographic maps.**

d) Please highlight in the conclusion (Line 341) that this analysis is for continental performance evaluation and may/may not apply to small regional or local domains.

We have added text to reflect your suggestion. Current Line 448-450: 'As with any continentalscale evaluation our results may (or may not) apply to small regions or local domains and the validation results do not apply to non-alpine areas which constitute a large proportion (~30%) (Wrzesien et al., 2019) of the total hemispheric SWE.'

**Technical corrections**

Abstract – A '0.1' increase in correlation does not seem very significant. Please provide justification of the significance of this increase in correlation using additional analysis (such as hypothesis testing). This

analysis can be included as a separate paragraph in Section 3.3. If no justification exists, then it would be advisable to remove the sentence from the abstract.

**Sentence revised and reflects addition of ERA5 and MERRA2:**

'Using a seven-dataset ensemble that excluded the standalone passive microwave products reduced the RMSE by 10 mm (~20%) and increased the correlation from 0.67 to 0.78;...'

Line 32 – Please define which seasonal forecasts the authors are eluding to. Being specific will make the discussion more accessible to the reader.

The statement on the verification of seasonal forecasts is followed by the reference to Sospedra-Alfonso et al. (2016). This citation provides details on the study which used various snow analyses to verify forecast from the Canadian Seasonal to Interannual Prediction System (CanSIPS).

Line 38 – Please define the difference between 'snow depth' and 'surface snowfall' measurements. 'Snow depth' refers to snow on the ground measured by a ruler, sonic snow depth instrument, etc.; 'surface snowfall' refers to snowfall measurements from a precipitation gauge. We removed the word 'surface' which clarifies the sentence (lines 37-39).

'Meaningful spatially continuous information can be derived from surface observations for regions and time periods with a sufficiently dense observing network (Dyer and Mote, 2006; Brown and Derksen, 2013); as an alternative to snow depth, snowfall measurements can also be integrated (Broxton et al., 2016).'

Line 40 – It isn't clear what 'coarse' grid cells means. Can you quantify what you mean by 'coarse'? Good point. We re-phrased this sentence (lines 40-43):

'However, both snow depth and snowfall measurements from single point locations are intrinsically limited by a lack of confidence in how they capture the landscape mean when gridded (Meromy et al., 2013), which is particularly problematic in areas of mixed forest vegetation, open areas prone to wind redistribution, and complex topography (most snow covered regions fall into at least one of these categories).'

Line 83 – Replace evaluation with evaluate. Change made. (Line 88)

Line 110 – Rephrase the sentence to highlight the difference between using separate brightness channels versus spectral difference for SWE estimation.

We feel it is too much technical detail to describe the difference between individual brightness temperature measurements and the spectral gradient. But we have modified the wording to clarify the sentence (line 124-126):

'The approach evolved from standalone passive microwave algorithms (it also relies on 19 and 37 GHz measurements), but the retrieval also integrates daily surface snow depth measurements.'

Line 162 – Please add an appropriate reference for the EASE2 grid.

Reference to Brodzik et al. 2012 added to line 210-212:

'For each temporal grouping, snow course measurements falling within a given 25 x 25 km EASE grid cell (Brodzik et al. 2012) were averaged together, thereby forming a gridded snow course field.'

**Added to reference list:**

Brodzik, M. J., Billingsley, B., Haran, T., Raup, B., & Savoie, M. H.: EASE-Grid 2.0: Incremental but significant improvements for Earth-gridded data sets, ISPRS International Journal of Geo-Information, 1, 32-45, https://doi.org/10.3390/ijgi1010032, 2012.

Line 220 – Please elaborate briefly on what the 'acceptable' uncertainty is for SWE. Please be specific in terms of quantitative (rather than qualitative) values of SWE uncertainty.

Added specific quantitative values. Revised (current) Line 317-318:

'It is important to note that the RMSE of the best performing products is at the margins of acceptable uncertainty for operational (<15%, Rott et al., 2010; Larue et al., 2017) and scientific (10-25%; Derksen and Nagler, 2019) requirements.'

Line 225 – GlobSnow also underestimates SWE above values > \_130mm. This statement needs to be included here.

Added statement regarding underestimation of GlobSnow above >130 mm, current Line 327: 'GlobSnow overestimates SWE up to ~100 mm and underestimates above ~130 mm ...'

Line 239 – Figure 3g shows >70mm rather than >60mm as the pivot point.

**Thank you for this observation, we have changed the text to reflect this. Current Line 229:** 'The AMSR-E v1.0 product exhibits low sensitivity to SWE, especially for values >70 mm and overestimates low SWE values.'

Line 266 – contain is written twice. Second 'contain' removed.

Figure – 1: Extra 'E' in AMSR-E in caption. Extra 'E' removed from AMSR-E in caption.

Figure – 3:

a) Please define how Figure 3 was developed in the main text. Is the binning based on average SWE values for each grid cell or the average of bi-weekly values for all snow courses?

We have added additional text to explain how the binning was conducted. Reviewer 1 asked that we reorganize the description of methods that were interspersed with the results. The clarification about Figure 3 (now Figure 4) was added to this revised methods section.

**Added to lines 229-231:** 'Finally, to determine the influence of SWE magnitude on product performance, all snow course-product SWE pairs were binned into 10 mm increments according to the gridded SWE estimate. For each 10 mm increment the average product SWE is plotted against the bin midpoint.'

b) The term 'retrieval' is used in the caption. This term does not apply to all the different SWE datasets being evaluated. Please change the sentence from 'retrieval performance versus reference SWE ....' to 'Performance of SWE datasets versus reference SWE measurements .....'

Thank you for this suggestion. We have revised Figure 3's (now Figure 4) caption accordingly: Figure 4. Performance of SWE datasets versus reference SWE  $\pm$  1 standard deviation for (a) Crocus; (b) ERA-Interim/Land; (c) ERA5; (d) GLDAS; (e) GlobSnow v2.0; (f) MERRA; (g) MERRA2; (h) AMSR-E v1.0; (i) AMSR-E v2.0. SWE values above 300 mm are not shown. c) The figure labels for subplots f and g do not match the caption labels.

We have revise this figure caption. See previous response.

Figure – 4: In the text, a bi-weekly time step is specified while the caption describes a ten day time step. Please clarify this contradiction.

Although the use of 10-day steps for Russia is described in the methods section, we have clarified the text (current Line 309-310) to specify that ten day time steps were used for Russia.

**Revised methods lines 226-227:**

To understand the influence of seasonality on product performance, bias, RMSE and correlation were also computed across all years for each biweekly period (10 day period for Russia). Results lines 338-339:

'To quantify the influence of seasonality on product performance, validation statistics (RMSE, bias, correlation) were computed at a bi-weekly time step (10-day for Russia) for 2002 through 2010 (Sec. 2.3).'

Figure – 6: The figure title can be removed since the caption and labels are self explanatory. **Figure title removed.**

Data availability – There seems to be a typing error in the data availability description. The NASA AMSR-E operational dataset is mentioned twice.

Thank you for catching this. The first mention of NASA operational should have read NASA historical. We have revised the nomenclature surrounding the NASA products. NASA Historical is now NASA AMSR-E SWE v1.0 and NASA Operational is now NASA AMSR-E SWE v2.0. Data availability now reads:

'Data availability. Météo-France provided data from the Crocus snowpack model; the NASA AMSR-E SWE v1.0 dataset and is available from the paper's authors upon request as is the NASA AMSR-E SWE v2.0 dataset. The remaining datasets are available for download via the links and references provided in Sect. 2.'

---

## Author Response (AR2)

**Response to Editor – minor revisions**

Thank you for the positive feedback regarding our revised manuscript. We agree that the comments from the reviewers greatly enhanced the manuscript. We have accepted the suggestion to move Figure S1 from supplementary material to the main text. Figure S1 is now Figure 7 (see below). Old figures 7 and 8 are now 8 and 9, respectively. In-line references to these figures have been changed accordingly.

[revised manuscript text omitted]